



# Ability of the e-TellTale sensor to detect flow features over wind turbine blades: flow stall/reattachment dynamics

Antoine Soulier[1,2], Caroline Braud[2], Dimitri Voisin[1], Bérengère Podvin[3]

1- Mer Agitée, Port-la-Forêt, 29940 La Forêt-Fouesnant

2- LHEEA (CNRS/ECN), Ecole Centrale Nantes 1, rue de la Noë, 44321 Nantes

3- LIMSI (CNRS), Campus Univ. bât. 507, Rue John Von Neumann, 91400 Orsay

**Correspondence:** Caroline Braud (caroline.braud@ec-nantes.fr)

**Abstract.** Monitoring the flow features over wind turbine blades is a challenging task that has become more and more crucial. This paper is devoted to demonstrate the ability of the e-TellTale sensor to detect the flow stall/reattachment dynamics over wind turbine blades. This sensor is made of a strip with a strain gauge sensor at its base. The velocity field was acquired using TR-PIV measurements over an oscillating 2D blade section equipped with an e-TellTale sensor. PIV images were post-

processed to detect movements of the strip, which was compared to movements of flow. Results show good agreement between the measured velocity field and movements of the strip regarding the stall/reattachment dynamics.

## 1 Introduction

Wind turbines are placed in the low layers of the atmospheric boundary layer where the wind is strongly influenced by the surface roughness and the thermal stability which create turbulence and vertical gradients of the wind (Emeis, 2018). The

rotor yaw and the blade pitch alignment within this highly unsteady wind inflow is a subject that is becoming more and more crucial with the rotor blade lengths that are increasingly long (107 m for the largest existing turbine: Haliade-X). Also, offshore turbines are arranged in an array layout and not just in-line, which induces additional sheared inflow conditions and additional small turbulent structures (Chamorro et al., 2012). This results in strong and local variations of speed and directions on the wind turbine rotor blades. These variations lead to unsteady aerodynamic effects with turbulent inflows responsible for more

than 65% of fatigue loads (Rezaeiha et al., 2017). To alleviate these loads, smartblades and/or fluidic actuators are nowadays considered (Pechlivanoglou, 2013; Jaunet and Braud, 2018; Batlle et al., 2017). For this last strategy or to perform blade remote monitoring, one key issue is the development of robust technologies able to provide an instantaneous detection of the state of the flow on the blade aerodynamic surface. On current operating wind turbines the wind is generally monitored using





an anemometer situated on the nacelle. It provides a slow measure of the wind which is perturbed by the rotor and the nacelle.
Moreover being only a one-point measurement, it does not appreciate shear, yaw/pitch misalignments or turbulence on blades.
Recent monitoring technologies allow to overcome some of these drawbacks. Among the most mature technologies, the spinner
anemometer is measuring the wind in front of the rotor, removing perturbations from the rotor (Pedersen et al., 2007). Also,
capabilities, costs and integration of nacelle-mounted LIDAR, measuring the wind inflow few diameters upstream of the rotor,
have been significantly improved during the last decades (Aubrun et al., 2016) (Bossanyi et al., 2014). However, to the knowl-
edge of the authors, nothing is yet able to measure the state of the flow on current blades. Some field measurement campaigns
were punctually performed for research purposes using pressure probes around dedicated manufacturied blades.However the
potential for using these sensors in a day-to-day operation of wind turbines is weak. (Troldborg et al., 2013). Some solutions
were explored such as tufts or stall flags glued on the blade correlated with positions of the flow separation (Swytink-Binnema
and Johnson, 2016; Pedersen et al., 2017; Corten, 2001). However, these methods need a mounted camera on the turbine with
its associated drawbacks (fragility of the camera, vision at night ...).

An interesting alternative to these technologies is the electronic telltale sensor, developed by Mer Agitée[1]. It is composed
of a strip moving like a tuft but with a strain gauge encased in its base making it able to transmit the information directly
to any monitoring or control system through an embedded wireless electronic unit. It has been originally developed to detect
flow separation on sails of offshore racing sailing vessels and has been recently adapted for wind turbine blade monitoring.
Robustness and practical mounting issues were solved from industrial tests (figure 1a), while full scale tests of the device
were performed at high Reynolds numbers in the NSA wind tunnel facility of CSTB[2], to demonstrate the relation between the
e-Telltale sensor signal and the lift curve for static variations of the angle of incidence as can be seen in figure 1b (Soulier
et al., 2017). It was found in particular that a e-Telltale sensor located at the trailing edge of the profil with a sufficiently long
strip is able to detect both: the trailing edge separation and the stall phenomena. The present study is intended to study the
ability of the e-TellTale sensor to dynamically detect the apparition of stall or reattachment process and to distinguish one from
the other. For that purpose, experiments of a downscaled 2D blade section, oscillating around the stall angle, were performed
in the LHEEA aerodynamic wind tunnel, using Time Resolved PIV and different post-processing methods to extract the strip
position of the sensor in the flow field (vision algorithms) and to evaluate instants at which the stall/reattachement phenomena
occurs over the aerodynamic surface. The objective of the e-TellTale sensor is to detect the apparition of stall/reattachement
for real-time monitoring or control purposes. Therefore, the detection methods used to validate this sensor are preferably using
instantaneous criteria: an instantaneous evaluation of the sign of the tangential velocity, an instantaneous evaluation of the
wake width. Only one statistical approach is chosen (POD decomposition). The experimental set-up and the post-processing
methods are described in paragraph 2 and 3 respectively. Results are presented in the 4th paragraph including: a description of
the baseline flow (4.1), results of the different post-processing methods to detect the flow stall/reattachement phenomena (4.2),
results on the ability of the e-TellTale sensors to detect flow separation (4.3).

---

[1]https://www.meragitee.com/
[2]http://www.cstb.fr/fr/

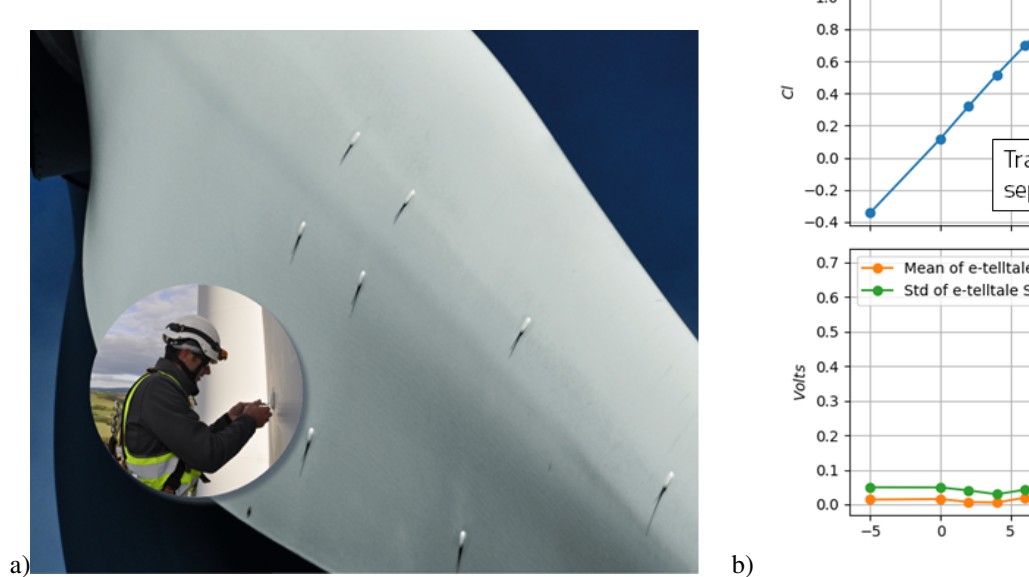

**Figure 1.** Previous studies: a) Robustness and practical mounting issues solve on EDF-Renewable wind turbines b) Ability of full scale e-TellTale sensors located at the blade trailing edge to detect static flow separations at high chord Reynolds numbers ($10^6$) from wind tunnel tests: increase of the eTellTale signal after stall angle, at 20°

## 2  Experimental Setup

The experiments were performed in the recirculating aerodynamic wind tunnel facility of the LHEEA laboratory at Centrale Nantes (France). The working section is 0.5x0.5 m² and 2.4 m long with a turbulent intensity less than 0.3 % of turbulence. The Reynolds number based on the chord length of the 2D blade section, $c \simeq 0.09\,m$, is $Re_c = (U_\infty c)/\nu \simeq 2.10^5$ with $U_\infty =$ 35 $m/s$ the free-stream velocity.

### 2.1  Blade profile

Measurements were performed using a NACA 65-421 profile in composite material. Due to the fabrication process, it is truncated at 91 % of the chord length so that the trailing edge thickness is 2 mm (see figure 2). A similar profile was already used by Jaunet & Braud(Jaunet and Braud, 2018) to demonstrate the ability of local micro-jets to alleviate loads. It is a thick profile with two drops on the lift coefficient curve corresponding to a first boundary layer separation at the trailing edge of the profile for $AoA \sim 8°$, and a second flow separation at the leading edge for$AoA \sim 20°$ causing stall. From 8° to 20° the separation point moves gradually from the trailing edge to the leading edge, corresponding to a gradual variation of the loads.



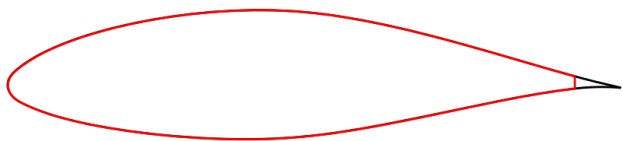

**Figure 2.** NACA $65_4$-421profile manufactured in red and the theoretical trailing edge in black

An oscillating motion was imposed using a crank drive for the linear movement imposed by a feedback linear motor from LinMot. This oscillating motion was checked from PIV image processing using the detection of the blade surface at the position of the e-Telltale sensor. The detection of the blade surface was also later used to extract the position of the e-TellTale sensor in the vector field (see section 3.1) and will give an information on the relative angle of incidence. The amplitude of the blade oscillation, $\Delta\alpha_0 = 5°$, was chosen so that the flow, initially separated at the trailing edge, moves gradually towards the leading edge flow separation where the stall occurs as it can be checked on PIV vector fields in figures 8 and 9. The oscillating frequency, $f_{osc} = 1\,Hz$, was chosen similar to the study of Jaunet & Braud (Jaunet and Braud, 2018) to mimic a constant shear inflow. This leads to a reduce frequency of $k = \pi f_{osc}c/U_\infty = 0.008$ corresponding to a quasi-steady stall behavior (Choudhry et al., 2014). The blade was equipped with a e-Telltale sensor at mid-span on the suction side. Figure 3b) shows the e-Telltale on the surface of the 2D blade profile installed in the LHEEA aerodynamic wind tunnel. A small part ($\simeq 5\,mm$) of the pink strip of the e-telltale sensor is glued on a strain gauge sensor, itself glued on a thin stainless steel sheet embedded into the blade. The rest of the strip is free to move above the aerodynamic surface. Its length is one third of the blade chord. The signal from the strain gauge sensor was not acquired simultaneously during PIV measurements, however, it has been checked before experiments that the signal from this strip, made of a nylon fabric, behaves similarly as full-scale experiments from (Soulier et al., 2017). In particular it was checked that it was possible to distinguish two levels of the signal within the blade oscillation cycle, corresponding to two different flow states over the aerodynamic surface: attached/stalled. Also, no load measurements were performed during PIV measurements, thus only the spatio-temporal informations will be used latter to detect the stall state on the aerodynamic surface.

## 2.2 PIV measurements

Flow data were collected with a TR-PIV system able to produce 1600 velocity fields each second. A DM20-527 DH laser from Photonics Industries delivering a 2x20 mJ double laser sheet at the green wavelength of 527 nm was used in this setup. The camera was a Phantom Miro M310, recording 1200 x 800 px² images at 3200 Hz, the 6 Gb of Ram memory of the camera allowed to capture 2000 velocity fields for each run. The camera was equipped with a Zeiss Makro Planar 2/50 lens (i.e. $f = 50mm$, $a = f/2$). With this setup, the field of view was 216 x 106 mm² leading to a spatial resolution of 6.3 px/mm. The PIV velocity fields were computed using a 16 x 16 px² interrogation area with an overlap of 50% leading to 159 x 99 vectors with a maximum spacing between vectors of 1.3mm or 0.014c. As seen in the figure 3 the optical axis of the camera





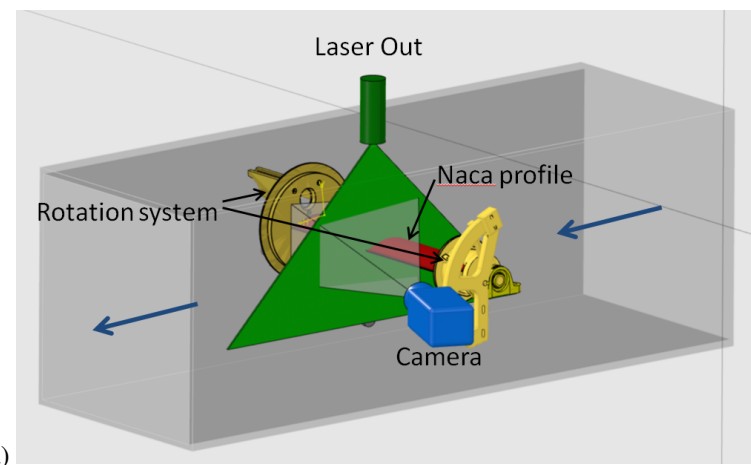

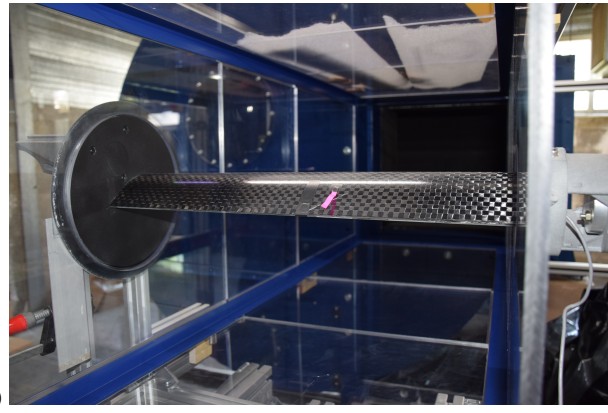

a)                                                                                          b)

**Figure 3.** Experimentat set-up in the LHEEA wind tunnel: a) scheme of the PIV set-up with the framework of the axis (x,y,z). b) the 2D blade section mounted in the test section with the e-telltale in pink

was not totally perpendicular to the laser sheet. After calibrating this misalignment by taking snapshots of a calibration target located at the measurement plane, all the raw images and the velocity fields were dewarped. In addition to the classical noise inherent to PIV measurement, the presence of the e-Telltale strip in the field of view of the PIV camera caused some spurious vectors explained by some light shoots on images when the clear fabric of the strip reflect the laser light directly towards the camera. To remove and replace these spurious vectors the automated post-processing algorithm developed by Garcia (Garcia,

2011) was used.

## 3    Introduction in processing methods

### 3.1    Strip detection method

The flow field over the aerodynamic surface is measured using TR-PIV measurements during the oscillations of the blade profile. To extract movements of the e-TellTale strip within this flow field, PIV images were post-processed using vision

algorithms from the Open Source Computer Vision Library[3]. The chosen methodology uses PIV images containing laser reflections of the blade surface and of the strip. The first step is to separate the blade surface contour from the strip contour. The images were first binarized so that white pixels, corresponding to the reflection of the laser on the blade and the strip surfaces, are set to 1 and all others to 0. To separate pixel coordinates of the blade from pixel coordinates of the strip, a local gradient of white pixel coordinates is computed, revealing ordinates of pixels corresponding to the strip location. Then, the resulting curve

was smoothed using a Savitzky-Golay filter. Finally, this resulting identified profile curve was fit to the theoretical suction side profile curve to extract the best euclidean transformation (i.e. only rotation, translation and uniform scaling considered for the transformation) going from the measured curve to the theoretical profile. This was done using a function of OpenCV

---

[3]http://opencv.org





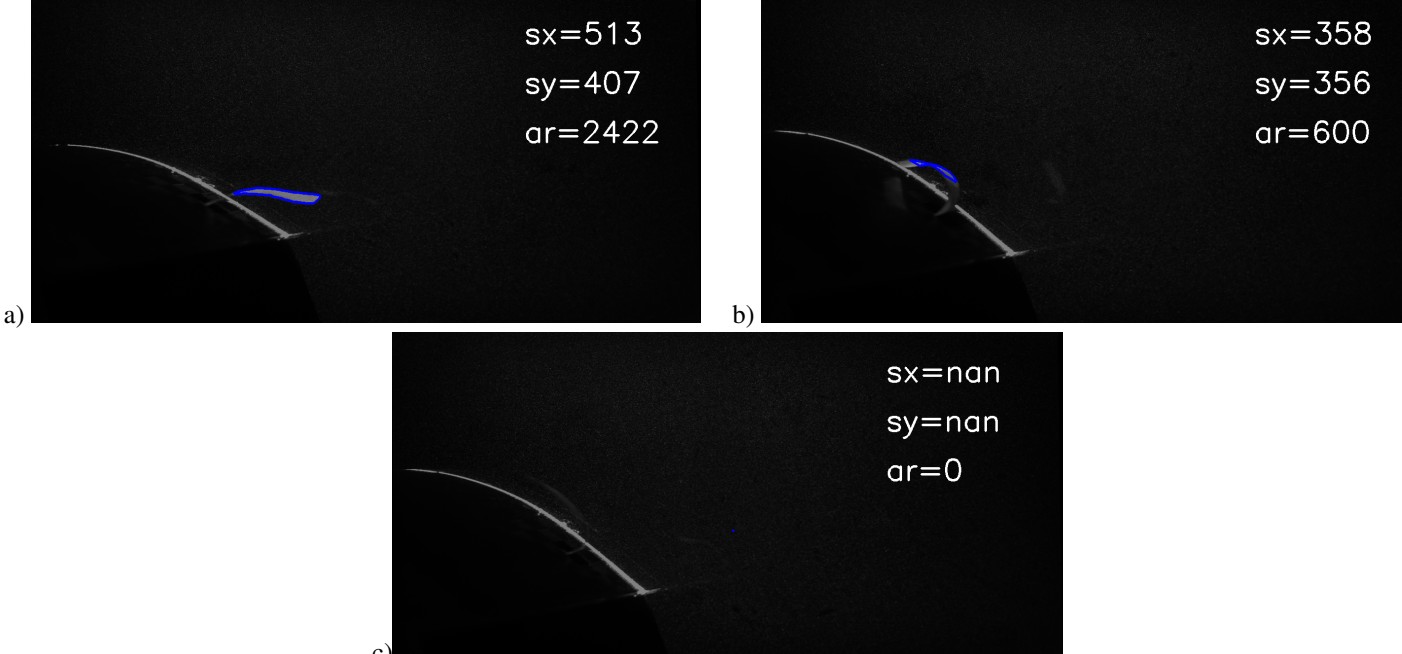

**Figure 4.** Detected strip contour from PIV images using OpenCV: a) for the attached flow case and b) for the detached flow case c) corresponding to an outlier case (impossible to detect the strip position). sx and sy are respectively the streamwise and spanwise positions in pixels of the center of the detected area (in blue). ar is the area of the detected contour in pixels

which primarily uses the RANSAC algorithm to detect spurious points and then the Levenberg-Marquardt algorithm to fit the profile. The result is a transformation matrix from which an angle of rotation is extracted. Also, from the detected blade surface
contour, a mask is defined to remove everything below it so that the remaining bright contour is the strip. The resulting cleaned binarized images were then used to extract the strip location using a contour detection function from OpenCV. The contour detection function recognizes the white pixels surrounded by other white pixels and regroups all of it in one entity. As we are interested in the flow separation phenomena over the aerodynamic surface which induces large movements of the strip from the downstream to the upstream flow direction, it was found sufficient to summarize the position of the strip by the center position
of the detected contour. The strip detection method was first validated on some samples such as the figure 4 which shows raw PIV images on which the detected aera is circled in blue with the coordinate of its center noted $sx$ and $sy$ for the respective streamwise and spanwise directions. It was then possible to automatize the method for images of the oscillating blade periods. Missing values present in the signal are related to default in the contour detection algorithm as can be seen in the figure 4c. These outliers are found to be correlated with AoA beyond stall, were 3D effects are dominants. These values were replaced
by the maximum value of $sx$. The corrected signal, $sxc$, is presented with the original signal $sx$ in the figure 5.

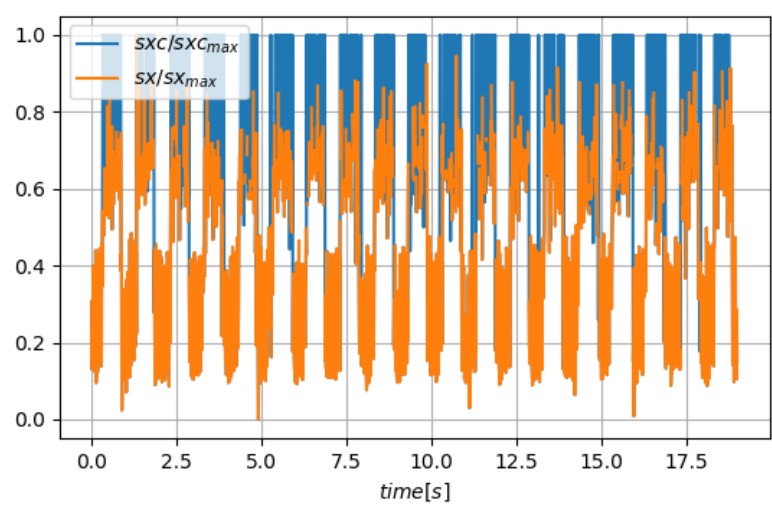

a)

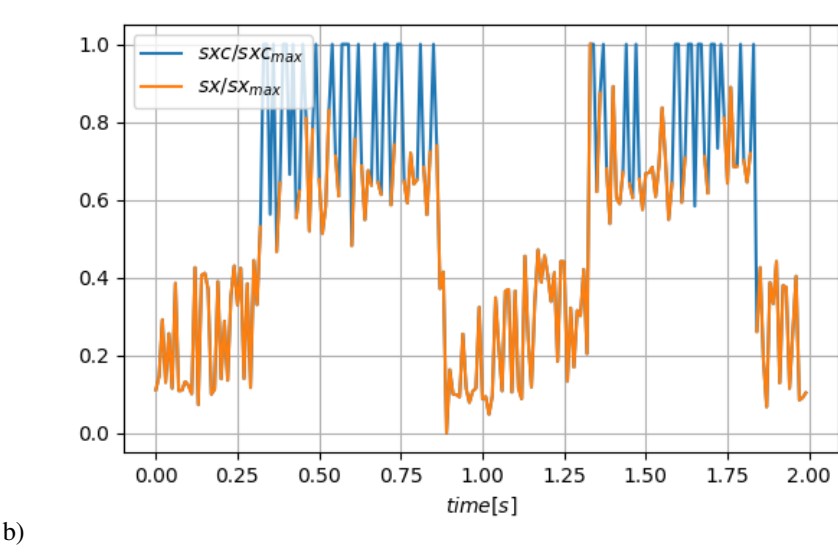

b)

**Figure 5.** Streamwise coordinate of the identified strip $sx$ before correction and $sxc$ after correction a) the full run and b) a zoom on the two first oscillations



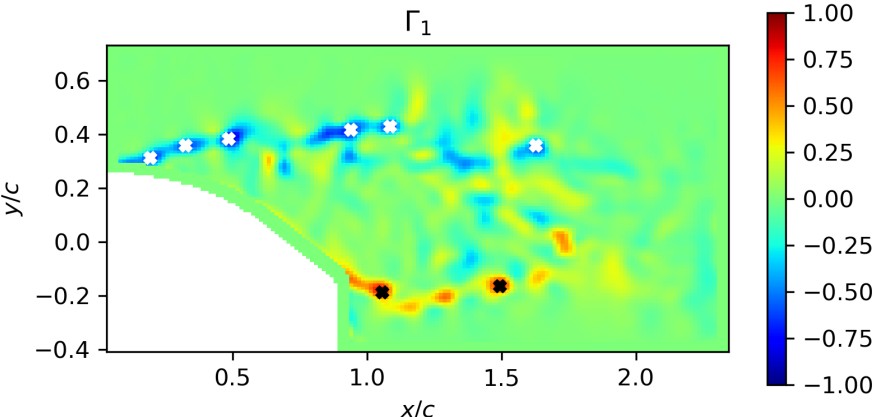

**Figure 6.** Exemple of an instantaneous isocontour map of the $\Gamma_1$ field with peaks identified using white cross markers for clockwise vortices and black cross markers for anticlockwise vortices

### 3.2 Vortex identification method

Vortex identification methods are widely spread in the literature (see e.g. (Jeong and Hussain, 1995)). As they enable to distinguish swirling motion from shearing motion, they were developed to help in the understanding of turbulent flows and more recently as a real-time processing method for flow control purposes (see e.g. (Braud and Liberzon, 2018)). In the present

study, the $\Gamma_1$ criterion method is used (Michard et al., 1997). This is a geometrical criterion defined as follows:

$$\Gamma_1(P) = \frac{1}{N} \sum_S \frac{(PM \wedge U_M).z}{\|PM\|.\|U_M\|} \tag{1}$$

where $N$ is the number of $M$ points of the square area $S$ around the center $P$, $U_M$ the velocity at the point $M \in S$ and $z$ the normal unit vector. The size of $S$ acts as a spatial filter. For this study different sizes of $S$ from 9 to 3 grid points were tested and the differences were found not significant. The presented results were obtained with $S$ being a square of 7 points.

From this definition, $\Gamma_1$ is a dimensionless scalar ranging from $-1$ to $1$, which local extremum indicates the center of a vortex. Compared to other methods such as the well known Q criteria, the $\Gamma_1$ criteria provides equivalent results, with the advantages to avoid computation of gradients (i.e. decreasing noise sensitivity) and to provide the sign of vortices. Similarly as (Mulleners and Raffel, 2013), the vortex identification method was used to extract vortex locations in the shear areas over the blade surface during the blade oscillation cycles (see figure 6 for an illustration of an instantaneous $\Gamma_1$ field).





### 3.3 Proper Orthogonal Decomposition

The Proper Orthogonal Decomposition (POD), is a statistical technique (Holmes et al., 1996) that extracts spatial modes $\underline{\Psi}(\underline{x})$ that are best correlated on average with a given field $\underline{u}(\underline{x},t) = (u,v)$ defined on a domain $\Omega$. Let $<.>$ denote the temporal average. The field $\underline{u}(\underline{x},t)$ can be written as a superposition of spatial modes whose amplitude varies in time

$$\underline{u}(\underline{x},t) = <\underline{u}(\underline{x},t)> + \sum_n a^n(t)\underline{\Psi}^n(\underline{x})$$

The modes can be identified with the method of snapshots (Sirovich, 1987), which is based on the computation of the temporal autocorrelation $C$ for a given set of $N$ snapshots $\underline{u}(\underline{x},t_i), i=1,\ldots N$:

$$C_{nm} = \int_\Omega \underline{\tilde{u}}(\underline{x},t_n)\underline{\tilde{u}}(\underline{x},t_m)d\underline{x},$$

where $\underline{u}$ represents the fluctuating part of the snapshots ($\underline{\tilde{u}}(\underline{x},t_n) = \underline{u}(\underline{x},t_n) - <\underline{u}(\underline{x},t)>$). The temporal amplitudes are eigenfunctions of

$$C_{nj}a^p(t_j) = \lambda^p a^p(t_n)$$

They are uncorrelated and their variance is given by

$$<a^n a^m> = \lambda^n \delta_{nm}.$$

The spatial modes are then obtained from

$$\underline{\Psi}^n(\underline{x}) = \sum_{i=1}^N a^n(t_i)\underline{u}(\underline{x},t_i).$$

By construction, the modes are orthonormal

$$\int_\Omega \underline{\Psi}^n(\underline{x}).\underline{\Psi}^m(\underline{x})d\underline{x} = \delta_{nm}.$$

POD was applied to the 2-D PIV vector fields over two different domains. The largest domain is used in the description of the baseline flow (section 4.1), while the smaller domain is used to detect the flow stall/reattachment dynamics in the oscillating cycle (see section 3.1).



## 4 Results

Results are presented in three steps. Firstly, the baseline flow obtained with an oscillation frequency of the blade $f_{osc} = 1\,Hz$ and an acquisition frequency $f_{PIV} = 100\,Hz$ is described, including a description of the flow during an oscillation cycle and the description of the secondary oscillation in the wake flow when separated. From this PIV field visualization, a first evaluation of the stall/reattachement instants is performed and called the visual reference. Secondly, three methods to detect the flow stall/reattachement instants from PIV measurements are presented and compared. Thirdly, results of the detection of the strip are compared to all detection methods to evaluate the ability of sensor to detect the flow stall/reattachement dynamic.

### 4.1 The baseline flow

One period of the blade oscillation relative angle, $\Delta\alpha$, is extracted using the blade contour mask from PIV images as explained in section 3.1 (see figure 7). The time duration $T$ and the amplitude of the blade oscillation were chosen to include the flow separation phenomena for quasi-static stall conditions, as previously described in section 2.1. Points of interest within this oscillating period are marked with letters from (a) to (i) and the corresponding instantaneous vector fields are presented in figures 8 and 9. At the beginning of the oscillating period, $\Delta\alpha = 0°$ and $t/T = 0$, the flow is slightly separated at the trailing edge of the profile as can be seen in figure 8a. From point (a) to (c), corresponding to a positive blade incidence variation, the separation point moves gradually from the trailing edge to the leading edge of the profile and the wake width increases accordingly as illustrated from 8a to 8b. From point (c) to point (d) the separation point suddenly moves towards the leading edge with a corresponding brutal increase of the wake width, until the flow is fully separated over the aerodynamic profile (see figure 8c and d). This last phenomena is ten times faster than the previous one and is clearly related to the stall phenomena. From point (d) to point (e), the flow downstream the blade can clearly be considered as an assymetric wake flow with shear layers on both sides of the blade (see figure 8d and e).

From point (e) to (g), despite the progressive decrease of the adverse pressure gradient on the suction side of the blade through a negative variation of the blade incidence during 0.3 seconds, the flow remains fully separated (see figure 9 e, f and g). From point (g) to point (h), corresponding to a duration of $\Delta t = 0.02s$, the separation point suddenly moves back towards the trailing edge. Again, this phenomena is ten times faster than the time duration from (e) to (g) for which the blade incidence is progressively decreasing (see figure 9 g and h). From point (h) to (i), the separation point is back to its initial state (see figure 9 h and i). This is the first visual method to detect the stall and reattachment instants, defined respectively as $t_{stall}^{ref}(ic) = (t_c + t_d)/2$ and $t_{attach}^{ref}(ic) = (t_g + t_h)/2$ with $t_c, t_d, t_g$ and $t_h$ the instants (c), (d), (g) and (h) extracted from $ic = 1$ to $N_{cycle}$, $N_{cycle} = 18$ being the total number of instantaneous oscillation cycles. They will be used in the following sections as a first comparison for the flow stall/reattachement detection methods of section 4.2.

It should be emphasized that the stall/reattachment phenomena has a time scale corresponding to $\sim 10c/U_\infty$ in good agreement with the theoretical work of Jones (Jones, 1940), with a stall/reattachement location occuring within one third of the blade chord from the leading edge.

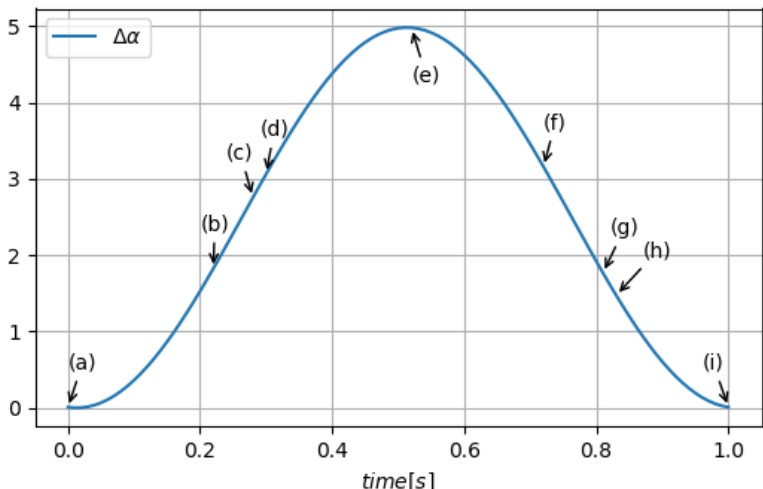

**Figure 7.** Evolution of the relative angle of attack $\Delta\alpha$. (x) : instantaneous velocity fields detailed below

To characterize further the coherent structure organization during this blade oscillation cycle, a POD analysis is performed from a database coming from a higher PIV acquisition rate, $f_{PIV} = 1600\ Hz$. All vector fields of the blade oscillation cycles

are used for the computation of the temporal autocorrelation coefficient $C$ (see section 3.3), corresponding to 2000 snapshots. The convergence of the resulting POD decomposition, in term of the relative energy content with modes, is presented in figure 10 using the following definition :

$$\Lambda_i = \frac{\lambda_i}{\sum_{j=1}^{N} \lambda_j}$$

where $N$ is the number of modes and $\lambda_i$ the eigenvalue of the $i$th-mode.

As highlighted from figure 10, the dominant modes in term of energy content are the three first POD modes, with around 14% of kinetic turbulent energy for the first mode, 10 % for mode 2 and 8 % for mode 3. These three modes are represented in figure 11 using the spatial modes, $\underline{\Psi}^n(\underline{x})$ with $n = 1, 2, 3$, together with the temporal modes scaled with the associated energy content, $a^n(t)/(2\lambda^n)$ with $n = 1, 2, 3$. The first mode is phased with the blade oscillation period and clearly captures variations of the mean velocity deficit in the wake due to these oscillations. The second and third modes exhibit structures in the wake

which could be associated to the vortex shedding organization, typically found in the wakes of bluff bodies. Following the work of (Yarusevych et al., 2009), the Strouhal number $St = f_s d/U_\infty \sim 0.22$ is extracted, with $f_s$ the peak frequencies from the FFT of temporal modes, $a^n(t)/(2\lambda^n)$ with $n = 2, 3$, and $d$ a measure of the wake width using the vertical distance between the two local maximum of the $r.m.s$ of the streamwise velocity at $x/c = 1.25$ . This Strouhal number is of the same order of



**Figure 8.** Instantaneous velocity fields superposed with isocontours of the velocity modulus (i.e. $\sqrt{\|u\|^2 + \|v\|^2}$) at different $\Delta\alpha$ corresponding to points of the blade oscillation given in figure 7 during the upstroke phase (noted ↗): (a) is a point at the lowest $\Delta\alpha$ of the upstroke phase of the oscillation cycle, (b) is an intermediate point, (c) is a point just prior to stall , (d) is a point just after the stall and (e) corresponds to a point at the maximum amplitude of the blade oscillation cycle



**Figure 9.** Instantaneous velocity fields superposed with isocontours of the velocity modulus (i.e. $\sqrt{\parallel u \parallel^2 + \parallel v \parallel^2}$ ) at different $\Delta\alpha$ corresponding to points of the blade oscillation given in figure 7 during the downstroke phase (noted ↘): (e) corresponds to a point at the maximum amplitude of the blade oscillation cycle, (f) is an intermediate point, (g) is a point just prior to the flow reattachment, (h) is a point just after the flow reattachment and (i) is a point at the lowest $\Delta\alpha$ of the downstroke phase





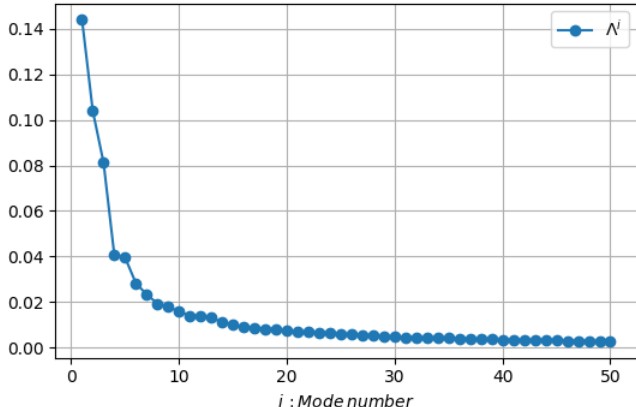

**Figure 10.** Energy content of each of the first 50 POD modes

magnitude that the one found by (Yarusevych et al., 2009) behind the wake of a NACA 0025 airfoil at the angle of attack of

10° and clearly assess the link of these modes to the vortex shedding organization behind the blade wake (see figure 12).

## 4.2  Detection methods

To be able to study the ability of the strip to detect the instants of the flow stall/reattachment phenomena, three robust detection methods were applied to the flow field obtained from the TR-PIV measurements:

  – Method 1: using the sign of the instantaneous tangential velocity component in the direction perpendicular to the surface

as introduced by (De Gregorio et al., 2007)

  – Method 2: using the instantaneous detection of the wake width from extraction of vortices in the shear layers as explained

      in section 3.2

  – Method 3: using the first mode of the POD decomposition introduced in section 3.3

In the perspective of using these sensors for real time control/monitoring purposes, the application of methods 1 and 2 to the

instantaneous PIV vector fields is preferred.

### 4.2.1  Method 1

For the first method, the apparition of stall/reattachment phenomena is detected using the normal profile of the instantaneous streamwise velocity component at a position corresponding to the attached strip location $x_{strip} = x/c \simeq 0.7$, $U_{norm}(t_i, x_{strip}, y_b)$ with $i$ the number of snapshots and $y_b$ the direction normal to the blade surface. The chosen line location is presented in white

on the figure 13. The normal profile is then reduced to a single value by averaging in the normal direction, $U_{norm}(t, x_{strip}) = \int_{y_b=0}^{l/c} U_{norm}(t, x_{strip}, y_b) dy_b$ with $l/c$ the normalized integration length in the normal direction, chosen so that each instant (or each angle of incidence) corresponds to one value of this normal velocity.



**Figure 11.** POD decomposition: a), b) and c) represents the eigenvectors vector field, $\Psi_i^n, i = 1, 2$ , of the first three modes respectively $(n = 1, 2, 3)$ with isocontours of its modulus superimposed, the associated energy content of the $n$ th mode (i.e. $\Lambda^n$) being written in the title, d) represents the corresponding temporal coefficients scaled with their energy content

### *Phase averaged approach*

The phase averaged of the obtained $U_{norm}(t)$ signal, $\overline{U_{norm}}$, is presented in figure 14a for different values of $l/c$ together with its gradient and with hatched time windows which width, marked by green and red hatched areas, corresponds to the standard deviation $\sigma(t^{ref}_{(stall-or-attach)}(ic) - ic.T)$ centered on the average of the reference instants extracted from the visualisation of instantaneous velocity fields of section 4.1, $\overline{t^{ref}_{stall}(ic)}$ and $\overline{t^{ref}_{attach}(ic)}$.

For low angles of incidence $\overline{U_{norm}}/U_\infty \simeq 1$, meaning $\overline{U_{norm}}$ is close to the free-stream velocity which corresponds to an

attached flow state over the aerodynamic surface. Similarly, for the large angles of incidence, $\overline{U_{norm}}/U_\infty$ is negative, bringing to light the reverse flow above the profile and thus the flow separation state. Between $\overline{U_{norm}}/U_\infty \simeq 1$ and $\overline{U_{norm}}/U_\infty \simeq 0$, the instantaneous vertical profiles contains reverse flow but not enough in average to be fully separated. The flow can be considered stalled or reattached if $\overline{U_{norm}}/U_\infty$ present a sudden gradient. Interestingly, whatever the value of $l/c$, the location



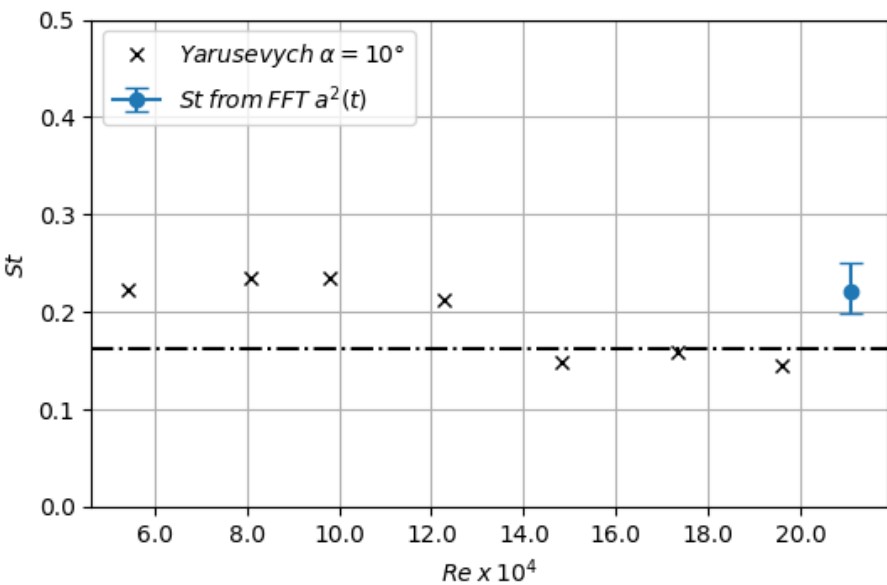

**Figure 12.** Strouhal number values extracted from (Yarusevych et al., 2009) and from the FFT of the temporal mode, $a^2(t)$, of the POD decomposition

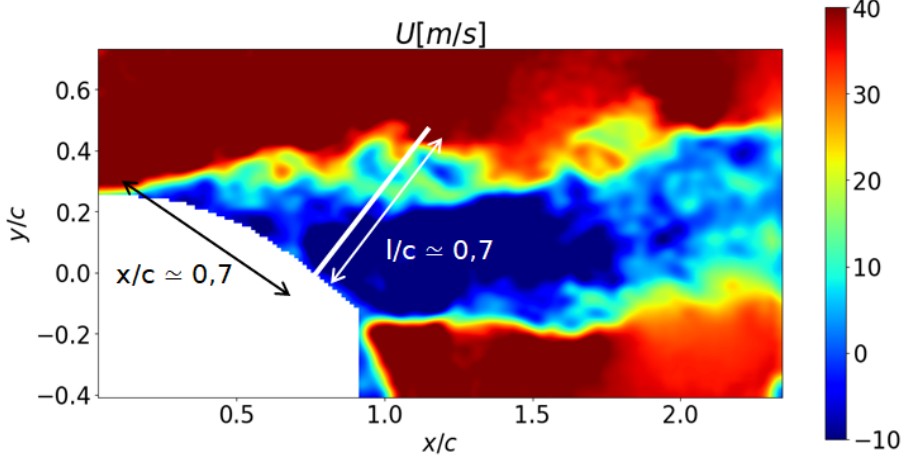

**Figure 13.** First method to detect the flow stall/reattachment instants: location and direction of integration line used to compute $U_{norm}(t)$ (i.e. white bar on the blade) reported on isocontours of the velocity modulus from PIV measurements

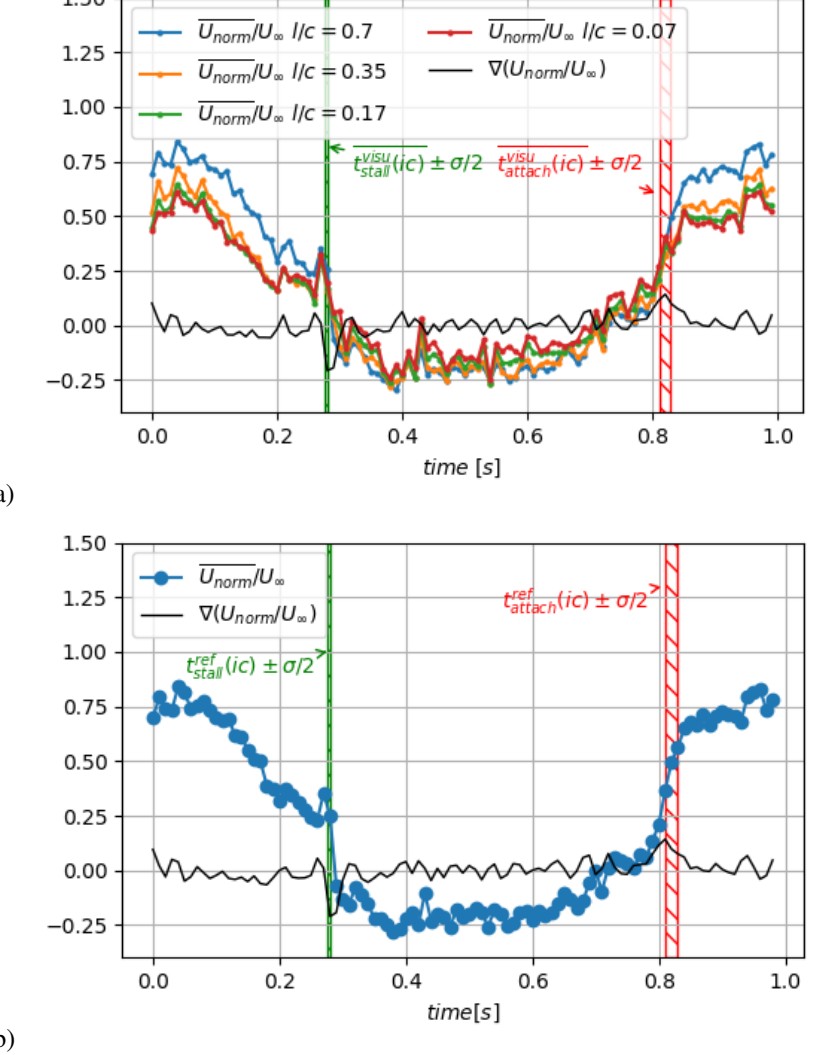

**Figure 14.** The phase-averaged signal of the first detection method $\overline{U_{norm}}$ with its gradient $\nabla \overline{U_{norm}}$ : a) for different value of $l/c$ b) for $l/c = 0.7$.



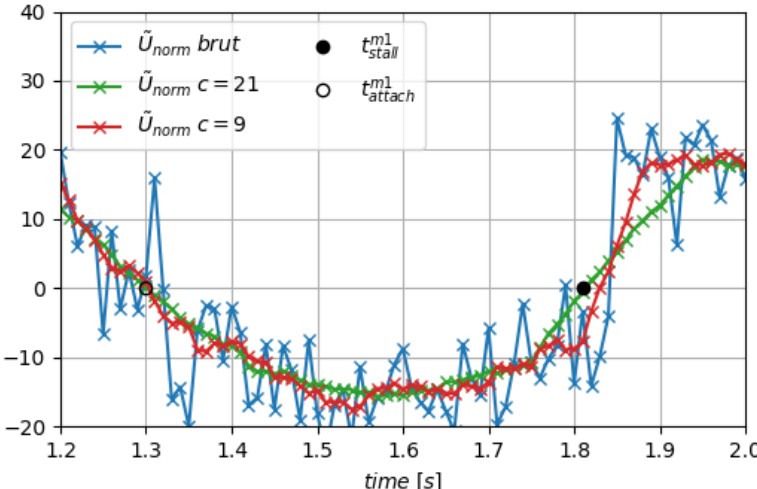

**Figure 15.** Zoom on a period of the instantaneous signal of $U_{norm}(t)$ (method 1), raw signal supperimposed with sliding averaged treatments with different filter width,

of the gradients is not modified. In the following, $l/c = 0.7$ is chosen (see figure 14a) which enables to have a higher amplitude

and thus a better signal to noise ratio to detect the gradients. Gradient peaks of the $\overline{U_{norm}}/U_{\infty}$ signal are close to these visual reference instants, whithin plus or minus one time step, which constitute a first validation of the method. It is interesting to note that the stall phenomena is marked by a rapid and strong modification of the $\overline{U_{norm}}/U_{\infty}$ value from 0.25 to 0, while the reattachement phenomena is smoother. This trend is also observed in other methods through a larger dispersion of the detected instants (see 1).


### Instantaneous approach

In wind tunnels, it is possible to reproduce known oscillations of the blade incidence to perform phase averaged treatments on signals as shown in figure 14. However, the targetted objective of e-TellTale sensors is to detect flow separations on operating wind turbines, without any inflow measurement. It is thus of interest to explore detection methods from the instantaneous

signals. As shown in figure 15, the raw signal $U_{norm}(t, x_{trip})$ needs to be smoothed in order to detect a unique stall and reattachement instant. Smooting instantaneous signals is a standard process to remove noise in real time control applications, however, in turbulent flow signals, this is equivalent to filter out the smallest turbulent structures and thus to obtain an ensemble average more or less biased (Cahuzac et al., 2010). The centered moving average algorithm is chosen here for its simplicity of implementation. From this treatment a filter size, which will be unique for sake of comparison with other detection methods,

needs to be defined. The main bias of this smoothing procedure is to reduce the gradient as illustrated in figure 15. Larger fiter size have a larger impact on the gradients, however, filter size as high as 21 time steps were found necessary to have an automatic procedure to extract stall and reattachement instants for all detection methods and thus having comparable results.

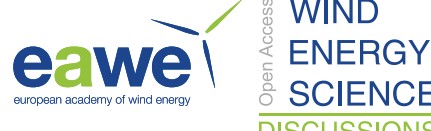

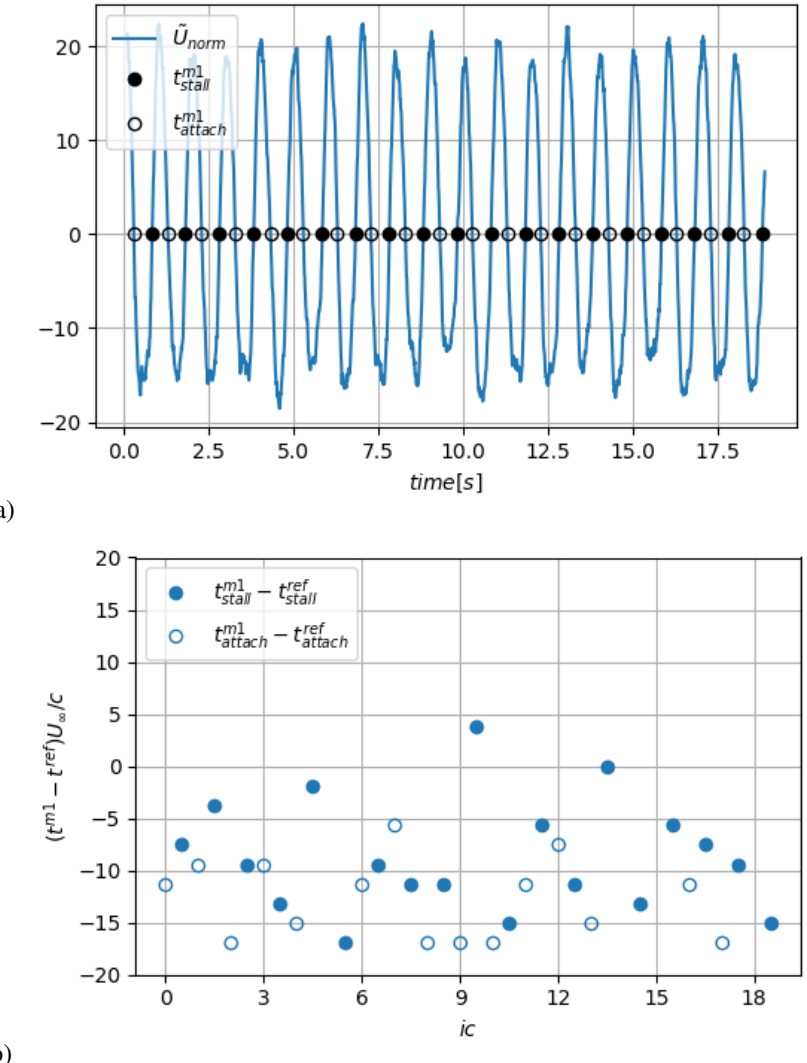

**Figure 16.** Results from the zero-crossing method applied on the $U_{norm}(t)$ signal: a) instantaneous signals with detected instants, b) Normalized delay of the stall and reattachment detected instants

Then, a zero-crossing criteria is applied to extract the detected instants $t^{m1}_{stall}(ic)$ and $t^{m1}_{attach}(ic)$. This criteria uses the $U_{norm}(t)$ signal minus its mean value, $\widetilde{U}_{norm}(t)$, so that, sudden variations of the signal are located where the fluctuating

signal is crossing the x-axis. Finally, the sign of the gradient, $sign(\nabla U_{norm})$, is used to discriminate stall instants from reattachment instants, $t^{m1}_{stall}(ic)$ and $t^{m1}_{attach}(ic)$ (see figure 16a). This zero-crossing method will be also used for the detection methods 2 and 3 that follows.

The resulting detected instants, $t^{m1}_{stall}(ic)$ and $t^{m1}_{attach}(ic)$ are compared to the reference instants extracted from the visualisation of the instantaneous velocity fields of section 4.1, $t^{ref}_{stall}(ic)$ and $t^{ref}_{attach}(ic)$ (see figure 16b). The first observation is that the





stall and reattachment instants are detected earlier in average than the visual reference, $\frac{\sum_{ic=1}^{N_{cycle}}(t_{stall}^{m1}(ic)-t_{stall}^{ref}(ic))}{N_{cycle}c/U_\infty} \sim \frac{-8.6c}{U_\infty}$ and

$\frac{\sum_{ic=1}^{N_{cycle}}(t_{attach}^{m1}(ic)-t_{attach}^{ref}(ic))}{N_{cycle}c/U_\infty} \sim \frac{-15c}{U_\infty}$ chord time respectively (or 2.5 to 4 time steps), which is to be related to the smothing

procedure used. Then, a certain dispersion exist in the detected instants that can be quantified using the standard deviations,

$\frac{\sigma(t_{stall}^{m1}(ic)-ic.T)}{c/U_\infty} = 3.3$ and $\frac{\sigma(t_{attach}^{m1}(ic)-ic.T)}{c/U_\infty} = 5.0$. Knowing the time resolution is $3.5U_\infty/c$, the same order of magnitude is

found for the visualreference ,$\frac{\sigma(t_{stall}^{ref}(ic)-ic.T)}{c/U_\infty} = 2.1$ and $\frac{\sigma(t_{attach}^{ref}(ic)-ic.T)}{c/U_\infty} = 7.0$. The higher dispersion in the reattachement

process is at the limit of the measurement precision. However, this trend is observed in the phase averaged signal (sharper

gradient), from visualisation of the flow (larger width of the hachted aeras) and also will be shown latter with method 2 and 3

(see table 1).

### 4.2.2 Method 2

Another flow separation detection method is introduced with, this time, a criteria associated to instantaneous vortices from

shear layers. Indeed,the vertical distance between identified vortices in the separated shear layers forming the blade wake

width is used, directly related to the flow separation location on the aerodynamic surface (Yarusevych et al., 2009) (see 3.2 on

the vortex identification method). The wake width is defined as :

$$W(t) = |\ \frac{1}{N_{clock}(t)} \sum_{n=1}^{N_{clock}(t)} y_n(t) - \frac{1}{N_{anti-clock}(t)} \sum_{m=1}^{N_{anti-clock}(t)} y_m(t)\ | \tag{2}$$

with subscripts $clock$ and $anti-clock$ corresponding to quantities from the clockwise and anti-clockwise rotating vortices

respectively and $N$ the number of vortices identified at the time $t$. The obtained signal can be phase averaged, $\overline{W(t)}$, as

presented in figure 17a. Gradient peaks of the $\overline{W(t)}$ signal are close to the reference instants, which standard deviation is

represented by green and red hatched areas. This constitutes a first validation of the method.

### *Instantaneous approach*

As for the first method, the zero-crossing criteria is applied to the resulting smoothed signal $W(t)$ to obtain stall and separated

instants for each instantaneous oscillating cycle, $t_{stall}^{m2}(ic)$ and $t_{attach}^{m2}(ic)$. First results show that the mean detected stall instant

is closer to the visual reference, i.e. $\frac{\sum_{ic=1}^{Ncycle}(t_{stall}^{m2}(ic)-t_{stall}^{ref}(ic))}{N_{cycle}c/U_\infty} \sim \frac{-2.5c}{U_\infty}$ (less than one time step), than the first detection method

presented in section 4.2.1. However, this is accompagnied by a high value of the standard deviation, $\frac{\sigma(t_{decro}^{m2}(ic)-ic.T)}{c/U_\infty} = 6.8$ ,

much more important than with other methods (see table 1). This augmentation is to be related to the difficulty to detect the

small vortices when shear layers are close to the blade surface and because their size this is of the order of magnitude of the

spatial resolution of PIV measurements. For the detection of stall instants, the method is more reliable as shear layers vortices

are bigger and futher away from the surface. However, the reattachment instants are detected significantly earlier in average

than the visual reference $\frac{\sum_{ic=1}^{Ncycle}(t_{attach}^{m2}(ic)-t_{attach}^{ref}(ic))}{N_{cycle}c/U_\infty} \sim \frac{-18c}{U_\infty}$ (~5 time steps: figure 17b) , due to the smoothing procedure

and similarly as method 1.



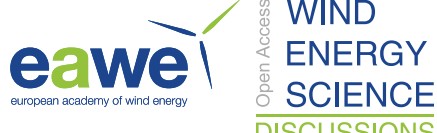

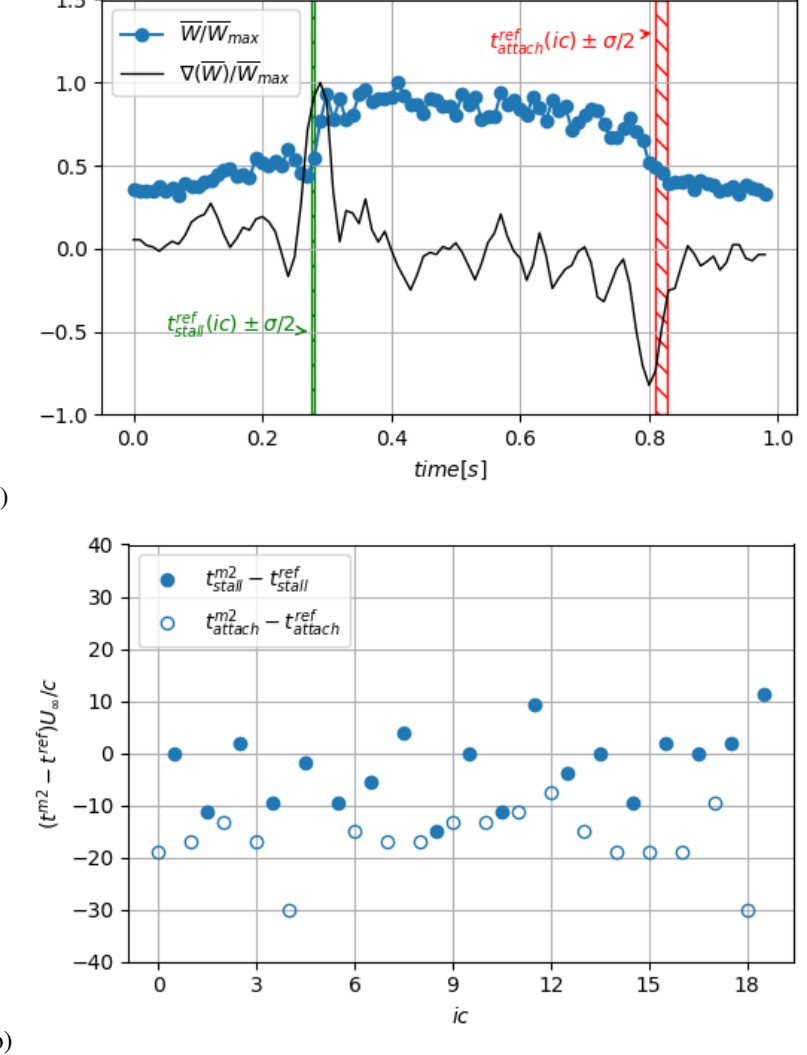

a)

b)

**Figure 17.** Second method to detect the flow stall/reattachment instants: a) The phase averaged signal $\overline{W(t)}$ with its gradient $\nabla \overline{W(t)}$, b) results of the zero-crossing method to extract the flow stall/reattachment instants using the second method, $t_{stall}^{m1}(ic)$ and $t_{attach}^{m1}(ic)$, compared to reference instants. The time window width marked by green and red hatched areas corresponds to the standard deviation $\sigma(t_{(stall-or-attach)}^{ref}(ic) - ic.T)$ centered on the averaged of the reference instants extracted from the instantaneous velocity fields of section 4.1, $\overline{t_{stall}^{ref}}(ic)$ or $\overline{t_{attach}^{ref}}(ic)$. The filled circle symbols correspond to stall instants, and void circle symbols corresponds to reattachment instants.





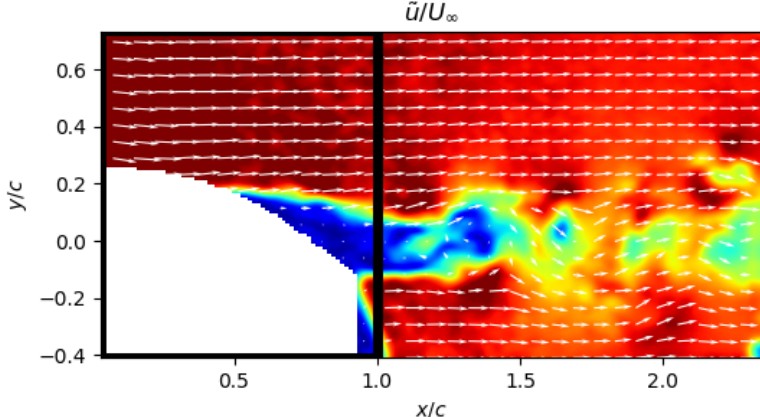

**Figure 18.** Reduced field of view (black rectangle) used for the third detection method using POD.

### 4.2.3 Method 3

These two previous methods provide an instantaneous detection of the flow stall/reattachement phenomena. To explore further the detection of these instants, we choose to use another method based on statistics introduced in section 3.3. It was already used in the context of wind energy for the analysis of the dynamic stall phenomena by (Melius et al., 2016; Mulleners and Raffel, 2012). The chosen vector field for the present analysis focuses on the separated shear layer dynamics rather than the wake dynamics from the initial PIV field of view (see figure 18). 2000 snapshots were used with no distinction of the phase, which enables to extract the flow separation state within the first POD modes as explained by (Melius et al., 2016; Mulleners and Raffel, 2012). As a first approach, the phase averaged of the two first POD modes are presented in figure 19 with temporal coefficients $a^1(t)$ and $a^2(t)$ . The first mode of the eigenvector field presented in figure 19a (i.e. $\Psi_i^1, i = 1, 2$), contains 77% of the total turbulent kinetic energy (i.e. $\Lambda^1 \sim 0.77$) and captures accelerations and deceleration of the flow over the profile depending on the sign of the associated temporal coefficient $a^1(t)$. The transitions between the accelerations (i.e. $a^1(t) < 0$) and deceleration (i.e. $a^1(t) > 0$) phases are marked by abrupt variations of amplitudes, which should be associated to instants of the stall and reattachment phenomena. The second mode of the eigenvector field presented in figure 19b (i.e. $\Psi_i^2, i = 1, 2$), contains much less turbulent kinetic energy (i.e. $\Lambda^2 \sim 0.049$ ) and exhibits a shear layer with a shear direction that is changing accordingly with the sign of its associated temporal coefficient $a^2(t)$. This variation of shear may be associated to the passage of the famous dynamic stall vortex created during unsteady variations of the angle of incidence as pointed out by (Melius et al., 2016; Mulleners and Raffel, 2012). Interestingly, minimums of $a^2(t)$ occurs significantly ahead of the flow stall/reattachment instants contrary to the first mode. However, studying the ability of the e-Telltale sensor to detect dynamic stall vortex needs further experimental investigations that won't be performed in this work. The following will therefore focus on the first POD mode.



**Figure 19.** Third method to detect the flow stall/reattachment instants: a) and b) are isocontours of the eigenvectors field $\Psi_i^n$, with $i = 1, 2$, of the $n$-th mode, with isocontours of its modulus superimposed. $\Lambda^n$ is the eigenvalue of the $n$-th mode, representing the part of the turbulent kinetic energy in the mode. c) represents the phase averaged of the corresponding temporal coefficients scaled with their turbulent kinetic energy content $(a^n(t)/\sqrt{2\lambda^n}, n = 1, 2,)$ . The time window width marked by green and red hatched areas corresponds to the standard deviation $\sigma(t_{(stall-or-attach)}^{ref}(ic) - ic.T)$ centered on the averaged of the reference instants extracted from the instantaneous velocity fields of section 4.1, $\overline{t_{stall}^{ref}(ic)}$ or $\overline{t_{attach}^{ref}(ic)}$.

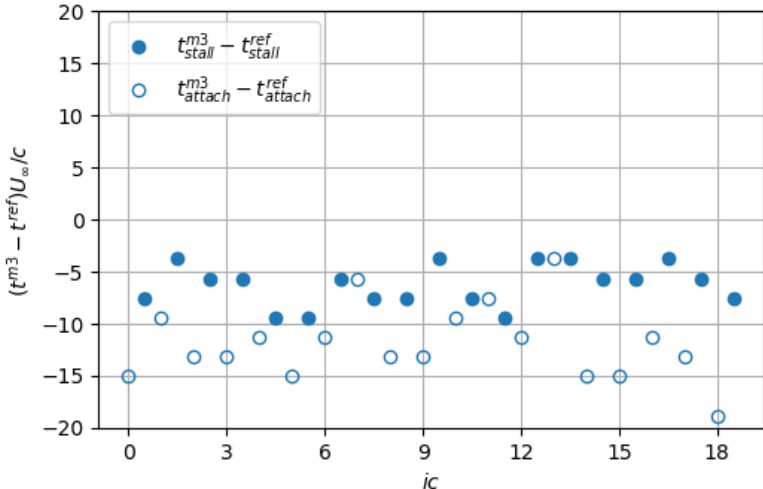

**Figure 20.** Results of the zero-crossing method to extract the flow stall/reattachment instants using the third method. The filled circle symbols correspond to stall instants, $t_{stall}^{m3}(ic)$, and void circle symbols corresponds to reattachment instants, $t_{attach}^{m3}(ic)$


### *Unsteady POD signal*

The coefficient of the first mode $a^1(t)$ was also studied instantaneously to compare with the other detection methods. A zero-crossing criteria was applied to this instantaneous signal, leading to detected stall and reattachment instants $t_{stall}^{m3}(ic)$ and
$t_{attach}^{m3}(ic)$. First results show that these instants follow the trend of the first detection method regarding the mean quantities, i.e. the detection occurs earlier than the visual reference similarly as method 1 and 2: $\sum_{ic=1}^{N_{cycle}}(t_{stall}^{m3}(ic) - t_{stall}^{ref}(ic)) \sim \frac{-6.2c}{U_\infty}$ and $\frac{\sum_{ic=1}^{N_{cycle}}(t_{stall}^{m3}(ic) - t_{stall}^{ref}(ic))}{U_\infty} \sim \frac{-12c}{U_\infty}$ , and the dispersion is also similar to the visual reference, i.e. $\frac{\sigma(t_{stall}^{m3}(ic) - ic.T)}{c/U_\infty} = 1.7$ and $\frac{\sigma(t_{attach}^{m3}(ic) - ic.T)}{c/U_\infty} = 5.7$. The delay is to be attributed to the smoothing procedure while the trend regarding the higher dispersion on the reattachement process is retrieved.

**4.2.4   Synthesis and comparison of detection methods**

Because the definition of stall and reattachement instants is a complex problem, four known or developed methods were used to detect both the stall and reattachement phenomena. Each method is using different spatio-temporal features of the flow over an oscillating profile, from the TRPIV measurements. As a first approach, the phase averaged signal is analyzed. All methods exhibit a phase averaged signal with the sudden variations associated to the stall and reattachment phenomena. Corresponding
instants can be extracted using a computation of the gradient, and the sign of the gradient enable to distinguish the stall from the reattachement phenomenon. The extracted stall and reattachement instants were found equivalent for all methods (within 1-2 time steps) to instants that can be extracted from the visual inspection of instantaneous vector fields, which provide a first





| | Mean of delays between stall detection with the different methods and the reference (a) | Mean of delays between reattachment detection with the different methods and the reference (b) | Standard deviation of the detected instants for stall (c) | Standard deviation of the detected instants for reattachment (d) |
|---|---|---|---|---|
| Reference | | | 2.1 | 7.0 |
| Method1 | -8.6 | -15 | 3.3 | 5.0 |
| Method2 | -2.5 | -18 | 6.8 | 7.8 |
| Method3 | -6.2 | -12 | 1.7 | 5.7 |

a) : $\frac{\sum_{ic=1}^{N_{cycle}}(t^{mj}_{stall}(ic)-t^{ref}_{stall}(ic))}{N_{cycle}c/U_\infty}$  b): $\frac{\sum_{ic=1}^{N_{cycle}}(t^{mj}_{attach}(ic)-t^{ref}_{attach}(ic))}{N_{cycle}c/U_\infty}$  c): $\frac{\sigma(t^{mj}_{stall}(ic)-ic.T)}{c/U_\infty}$  d): $\frac{\sigma(t^{mj}_{attach}(ic)-ic.T)}{c/U_\infty}$

**Table 1.** Summarize of detected stall/reattachement instants using the three methods including. All times are expressed as chord times ($t_c = c/U_\infty$)

validation of the methods. Then, targetting real time monotoring and/or real time control, the signals extracted from all methods were explored instantaneously. Method 1 and 3 were found to detect the stall and reattachement phenomenon similarly with an earlier (from 2.5 to 4 time steps) detection of the reattachement phenomenon compared to visual inspection of instantaneous vector fields. This early detection is to be attributed to the bias of the smoothing procedure on the gradients. Method 2 has a stall detection that occurs at similar values than what can be observed (visual reference) but with a larger standard deviation. This particularity is found to be related to the difficulties to detect vortices close to the blade surface. At last, all methods exhibits a larger duration and dispersion of the detected instants for the rattachement process, which is here highlighted as a particularity of this process.

### 4.3   Ability of the sensor to detect flow stall

Detection methods using TR-PIV measurements will be compared to the detection method using the e-TellTale sensor. For that purpose, the phase averaged strip position, $\overline{sx(t)}$, is detected from image processing as explained in section 3.1 and presented in figure 21 together with the time averaged standard deviation of stall and reattachment instants detected from the instantaneous flow field (i.e. green and red hatched areas respectively). It is observed that the position of the strip during the oscillation cycle is characterized by two sudden changes, revealed with the gradient peaks, in very good agreement with the stall and reattachment instants observed with the instantaneous flow field. This is a first validation of the e-TellTale sensor to detect stall and reattachment instants.

*Instantaneous approach*

To characterize further the detected instants from the movement of the strip, the zero-crossing criteria is applied to the corrected instantaneous signal of the position of the strip, $sxc(t)$. Resulting stall and reattachment instants removed by the reference instants, $t^{sxc}_{stall}(ic)-t^{ref}_{stall}(ic)$ and $t^{sxc}_{attach}(ic)-t^{ref}_{attach}(ic)$, are plotted in figure 22 and summarize in table 2. Contrary to what was found from other methods, the mean value is very close to the visual reference (i.e. close to zero). As can be seen

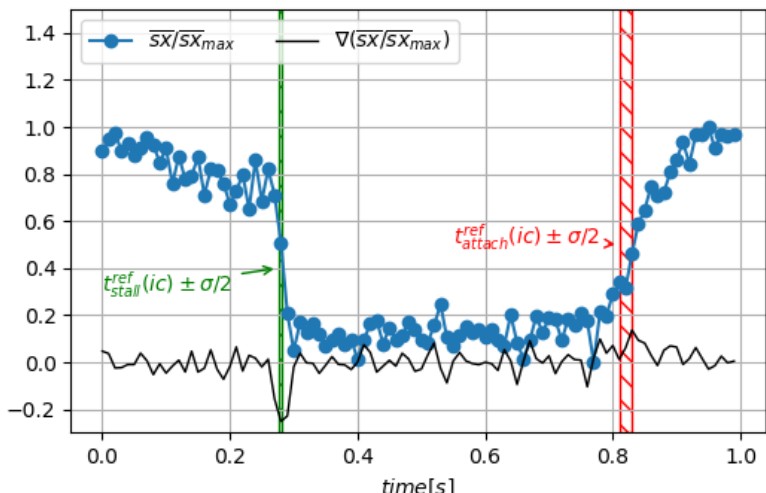

**Figure 21.** The evolution of dimensionless phase averaged streamwise coordinate of the center of the strip, $sx/sx_{max}$, during the oscillation cycle (blue dotted line) together with its gradient (black line) . The time window width marked by the red hatched area corresponds to the standard deviation $\sigma(t_{stall}^{ref}(ic) - ic.T)$ centered on the averaged $\overline{t_{stall}^{ref}(ic)}$ . The time window width marked by the green hatched area corresponds to the standard deviation value $\sigma(t_{attach}^{ref}(ic) - ic.T)$ centered on the phase averaged $\overline{t_{attach}^{ref}(ic)}$. .

| | Mean of delays between stall detection with the eTellTale method and the reference (a) | Mean of delays between reattachment detection with the eTellTale method and the reference (b) | Standard deviation of the detected instants for stall (c) | Standard deviation of the detected instants for reattachment (d) |
|---|---|---|---|---|
| E-Telltale | 1.1 | -1.2 | 8.7 | 5.2 |

a) : $\frac{\sum_{ic=1}^{N_{cycle}}(t_{stall}^{sxc}(ic) - t_{stall}^{ref}(ic))}{N_{cycle}c/U_\infty}$  b): $\frac{\sum_{ic=1}^{N_{cycle}}(t_{attach}^{sxc}(ic) - t_{attach}^{ref}(ic))}{N_{cycle}c/U_\infty}$  c): $\frac{\sigma(t_{stall}^{sxc}(ic) - ic.T)}{c/U_\infty}$  d): $\frac{\sigma(t_{attach}^{sxc}(ic) - ic.T)}{c/U_\infty}$

**Table 2.** Summarize of stall/reattachement detected instants using movements of the e-Telltale strip. All times are expressed as chord times $(t_c = c/U_\infty)$

on figure 23, this is related to the fact that the strong gradient is now centered around zero, due to the bias introduced with the correction applied on the original signal $sx(t)$ (5). The smoothing procedure has thus no effect on the detected instants. A particularity of the e-TellTale sensor, that remains an open question, is related to the standard deviation of the stall and reattachement process, respectivelly $\frac{\sigma(t_{stall}^{sxc}(ic) - ic.T)}{c/U_\infty} = 8.7$ et $\frac{\sigma(t_{attach}^{sxc}(ic) - ic.T)}{c/U_\infty} = 5.2$, which is more important for the stall than for the reattachement phenomena contrary to other detection methods. One hyppothesis is that fluctuations of the eTellTale strip movements are more sensitive to fluctuations of turbulent structures in the stalled configuration because they are larger and further away from the wall. This however needs further investigations at a higher acquisition rate.



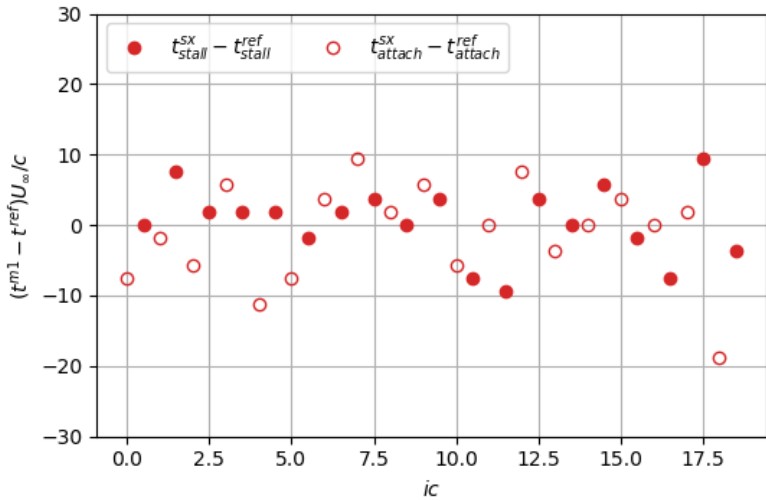

**Figure 22.** Comparison of the detected instants between the three methods using the instantaneous velocity fields and the position of the strip.

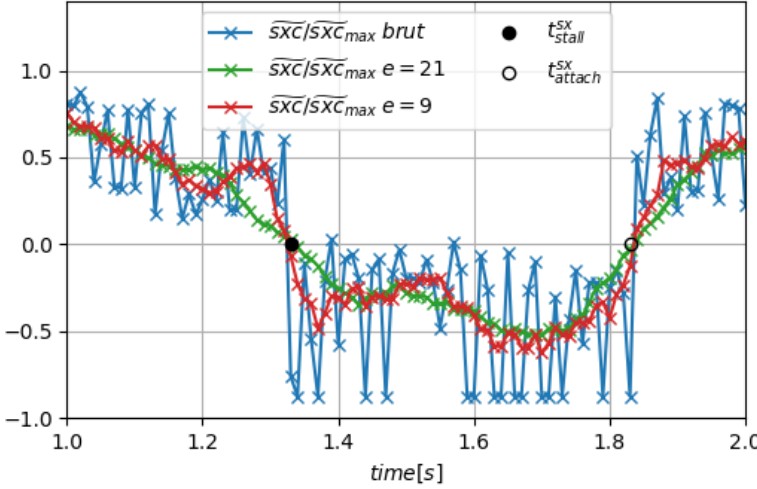

**Figure 23.** Zoom on a period of the corrected signal $\widetilde{sxc}(t)$ with the detected instants and the smoothed signal supperimposed using two filter size $e = 9$ et $e = 21$



# 5    Conclusion

The ability of an original e-TellTale sensor to detect flow stall/reattachment instants during oscillations of the angle of incidence of a blade section has been explored. For that purpose, a 2D NACA 65-421 blade section equipped with a e-TellTale sensor

at its trailing edge has been set in the LHEEA aerodynamic wind tunnel. The blade was oscillating around the stall angle to reproduce a constant shear inflow perturbations in front of a rotating wind turbine blade at a chord Reynolds number of $2.10^5$. Three methods to detect the flow stall/reattachment instants have been successfully applied using Time-Resolved-PIV measurements during the blade oscillation cycle. This includes two instantaneous methods: the direct use of the tangential instantaneous velocity (method 1) and the instantaneous extraction of shear layer vortices (method 2). One statistical method

is also tested using POD (method 3). Also, two types of treatments were applied on the extracted signal from the different methods: a phase averaged on the blade oscillating cycle and a direct use of the instantaneous signal.

The phase averaged signals of all methods give similar results of the detected stall and reattachement instants within 1-2 time step accuracy. Moreover, the sign of the gradient can be used to easily dicriminate the stall from the reattachement process.

The direct use of the instantaneous signals needs prior smoothing before applying the zero-crossing method to extract stall

and reattachement instants. Method 1 and 3 were found equivalent, with an earlier detection of the stall and the reattachment instants (2.5 to 4 time steps earlier), to be attributed to the smoothing method. Method 2, using an instantaneous detection of vortices, is not able to have an accurate detection of the stall instants due to the limitation of the spatial resolution close to the wall. However, reattachements instants were detected similarly as other methods. Also, all the detection methods, present a more sudden and less disperse stall phenomenon than the reattachement process

Then, results of these methods were compared to mouvement of the e-TellTale strip. The phase averaged signal of the strip movement is well correlated with all methods. This constitutes a first validation of the e-TellTale strip capabilities to follow the stall/reattachement dynamics. Also, when removing the smoothing effect, similar results were found regarding the mean values of the detected instants from an instantaneous processing of the strip signal, which constitute another validation of the ability of the e-TellTale strip to follow the stall/reattachement dynamics. An open question remains regarding fluctuations of

the strip motion which do not follow the trend found by other methods: higher fluctuations of the detected instants for the reattachement process than for stall process. Further investigations are needed with higher acquisition rate to investigate this higher order fluctuations of the strip position. In addition to the demonstration of e-TellTale ability to detect stall/reattachement instants, this paper introduce a methodology that could be used to evaluate the ability to extract other flow features of the blade aerodynamics such as the well known dynamic stall vortex or the blade wake dynamics. Also, what remains to be done is a

link between this dynamic strip position and the dynamic response of the e-TellTale strain gauge signal.

*Author contributions.*  Experiments were conceived and planned by C.B., A.S and D.V.; A.S. carried out experiments under the suppervision of C.B.; A.S. carried out the PIV post-processing; A.S. implemented the post-processing and detection methods (except the POD method) under the suppervision of C.B; A.S. implemented the POD method in collaboration with B.P. and under the suppervision of C.B.; Analysis were performed by A.S. under the suppervision of C.B., except for the POD detection methods which was discussed together with A.S, C.B.



and B.P.; A.S. wrote the first draft manuscript; reviews of manuscript were performed by C.B.; review of the manuscript regarding the POD part was performed by B.P.. A.S.'s PhD grant was obtained by D.V.. Other cost related to experiments were shared between C.B. and D.V. own funding.

*Competing interests.* The authors declare no conflic of interest.

**Acknowledgement**

Authors would like to thank Jean-Jacques Lasserre, Philippe Galtier for the PIV Dantec equipment loan and their assistance during measurements. We also would like to thanks Vincent Jaunet for his help during the PIV acquisition. This work was partly carried out within the framework of the WEAMEC, West Atlantic Marine Energy Community, and with funding from the city of Nantes, the Pays de la Loire Region and Centrale Nantes in France.



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
