# Peer review of "Ability of the e-TellTale sensor to detect flow features over wind turbine blades: flow stall/reattachment dynamics"

_Wind Energy Science, 2020_

## Referee Comment (RC1) · Anonymous Referee #1 · 26 Apr 2020

**General comments**

In this manuscript, the authors intend to show the ability of robust and practical electronic TellTale sensor to monitor the stall/reattachment of the flow on turbine blades in the field.

The wind tunnel experiment of 2D blade profile with 0.09m in chord length oscillated in 1Hz was conducted at Reynolds number of $2 \times 10^5$ with 100Hz TR-PIV measurement. The instants defined by the sudden change of the position of the sensor strip on suction surface detected by PIV image with vision algorism showed good agreement with exact stall/reattachment instants recognized from the velocity fields regarding the rapid

motion of the separation point. Other three postprocessing method to quantify the flow state are also discussed.

This work presents an important evaluation of the innovative device for the progress of the sophisticated turbine control including active flow control technologies on the blade. The various experimental techniques and the results to understand the characteristics of dynamic stall are very informative.

I strongly recommend this paper for publication with however revised extensively to clarify the importance of the work in the steps of realization of the technology to the real world and to focus the discussion of the result to the relation of the strip position to the exact stall/reattachment instants. I think it makes this work more impressive and helpful for the readers to understand the capability and challenges of this technology.

I hope the following comments help the authors for their revision.

**Specific comments**

1. The role of this work for realization of the device is not clear.

The importance of this work lies on the evaluation for e-TellTale but not for a tuft. It should be explained if there are any difficulties specific for e-TellTale to follow the flow dynamics, or to be recognized by image processing conducted in this work.

The most important feature of the sensor is the electrical sensing. But the electrical signals were not evaluated in this work. The correlation of the signals to the strip position should be described more in detail especially if there are some issues left.

If the authors intended to scale-down the full-scale device, the way of design to scale-down should be explained. The experimental condition or the configuration of the sensor for the full-scale wind tunnel test is not clear because the cited reference seems not yet published.

The TR-PIV is conducted in 2D. Does the 3D motion affect the electrical signals?

To think about this, it is recommended to describe more about the configuration of the e-TellTale in detail including the 'stainless sheet' and the 'small part'.

2. The validity of the position detection is not clear.

The position detection is the most important technique in this work. To ensure the validity of the experiment, clear and correct explanation is necessary. For example, why $sx$ replaced to $sxmax$ instead of $sxmin$ for the state beyond the stall in Fig.15 while $sx$ is decreasing when the flow is detached according to Fig.4.? Is the $sx$ really reaches to 0 at around 0.9s and 5.0s as shown in Fig.15 while the length of the strip is only 0.3c?

3. The objective and the result of the three postprocessing analysis is not clear.

The discussion about these analyses is too long and confusing while this manuscript is worthwhile enough for publishing even without these analyses.

'Because the definition of stall and reattachment instants is a complex problem' at l.321 is not clear to understand the objective because 'the definition' shown in section 4.1 is not complex.

If the objective of the analysis is to investigate the local flow phenomena which governs the motion of the strip, you might mention something more from the small l/c results of the method 1.

If the objective of the analysis is to evaluate the accuracy of each methods to detect the instants, the parameters for each method (such as $x/c$ or $l/c$ for the method 1) should be optimized before the comparison.

In section 5, there are no explanation that the exact instants $t^{ref}$ was defined by the visualization of the velocity field. Moreover, it is concluded that the strip capabilities to follow the stall/reattachment dynamics was validated by comparison to the three methods while the most direct validation seems to come from the comparison to $t^{ref}$. These are very confusing.

4. The validity of the zero-crossing criteria is not clear.

For about the 'resolution', describe the way of evaluation of 3.5c/U at l.262. Clarify the meaning of the phrase 'at the limit of the measurement precision' in l.265.

It should be described if there are reasons to set the detection threshold as zero. I think it should be optimized for each stall/reattachment instants for each method. Maybe this causes the 'bias' in l.350. Ideally, those instants should be compared to $t^{ref}$ after the optimization.

Moreover, if zero is calculated using the mean value in one cycle, the strategy on how to apply this to the field should be explained because the motion is not cyclic in the field.

The delay of the reattachment instances is described to be owing to the smoothing procedure in many sections. But I think the reason lies not only in the smoothing procedure but also in this threshold setting.

To think more about the interesting results that the dispersion of the delay is larger for reattachment than for stall, showing the average and the dispersion of the (td-tc) and the (th-tg) not only (tc+td) and (tg+th) is recommended to understand the rapidity of each phenomena.

**Technical corrections**

There are too many errors. Spelling should be checked. Symbols should be correctly defined. Adding the list of symbols is recommended. Labels should be added on each axis of the graphs. Too long paragraph should be divided. I'd like to ask authors for their comment if there are my misunderstanding

Section 1.
- l.31: The footnotes should be avoided in this journal. Web pages can be listed as references. Check the same trough the manuscript.
- l.39: The change of the paragraph is recommended before 'The present - '.

- l.47: The change of the paragraph is recommended before 'The experiment - '.
- l.48: paragraph -> section?
- Fig.1: solve -> solved?

Section 2.1
- l.54: Is '.' permitted in this journal? If not, check trough the manuscript.
- l.57: 65-421 -> 65(4)-421?
- l.68: Is it impossible to specify AOA not only $\Delta\alpha_0$?
- l.71: reduce frequency -> reduced frequency?
- l.72: a -> an?
- l.72: It seems not just at mid-span by Fig.3.
- l.76: Describe more about what and how for the checking.
- l.79: 'attached' means fully attached or TE separated?
- l.80: stall state -> stall/reattachment instance?

Section 2.2
- Fig.3: Experimentat -> Experimental
- Fig.3: Show the axis in the figure.

Section 3.1
- l.100: Add '(OpenCV)' for the later use.
- Fig.4: Does 'attached' mean fully attached or TE separated?
- l.117: spanwise -> vertical?

Section 3.2
- eq.1: Addition of an arrow over 'PM' is recommended to clarify the vector.
- l.127: M points -> points M?
- l.127: center P -> center point P?
- l.127: '∈S' is duplicated with 'points M of the square area'.
- l.128-129: How to place 3, 7, or 9 grid point in the square area around P?
- l.133: as (Mulleneres ans Raffel,2013) -> as the work by Mulleners (Mulleneres and

Raffel,2013) Is this manner permitted in this journal? If not, check the same through the manuscript.
- Fig.6: Exemple -> Example

Section 3.3
- l.139: Is the index $n$ of $a$ and $\Psi$ better to be written as $(n)$ to avoid confusing.
- l.142: $n$ is confusing because it is used for index of both eigen vectors and time.
- l.139: Equations should be numbered. Check the same through the manuscripts.
- l.143: Add  on $\underline{u}$

Section 4.1
- l.163: Is $\Delta\alpha$ same as $\Delta\alpha_0$ at l.68?
- l.175: How to know the pressure gradient?
- l.177: It is clearer to describe that (g) to (h) is related to reattachment phenomena as the same as l.172.
- l.183: Isn't it used for the second comparison not only for the first comparison?
- Fig.7: Add labels on the vertical axis or explain which is the direction of AOA increase.
- Fig.7: below?
- Fig.8: The label on the color-bar is not consistent with the figure caption.
- l.198: ( $2\lambda^n$ ) -> $\sqrt{(2\lambda^n)}$?
- Fig.10: Add a label on the vertical axis.
- Fig.11: The label on the color bar is not consistent with the figure caption.
- Fig.11: Add labels on each axis.
- Fig.11: How to mask the blade profile before the images with different AOA analyzed in POD?
- Fig.13: Add a label on the color bar.

Section 4.2
- l.207: What the 'robust' means?

Section 4.2.1

- l.218-221: Specify whether the x-y axis is moving with the oscillating blade or not.
- l.218: 'streamwise' means the direction of $U_\infty$?
- l.221: This integral doesn't represent an average.
- l.225: averaged -> average
- l.225: Unorm(t) -> Unorm (t, xstrip)
- l.226: Is 'gradient' refers to the spatial derivative? Otherwise, it is better to specify that it means time derivative. Check the same trough the manuscript.
- l.228 $t_{stall}^{ref}(ic)$ -> $t_{stall}^{ref}(ic) - (ic) \cdot T$. Check this trough the manuscript or, for example, define at l.181 as $t_{stall}^{ref}(ic) = (tc + td)/2 - (ic) \cdot T$ for the later simplicity.
- l.229: Is Unorm/U really nearly equal 1?
- l.234: sudden gradient -> peak of time derivative?
- l.234: Are these time derivatives for each l/s shown in Fig.14?
- l.234: 14a -> 14b?
- l.239: what is (see1)?
- Fig.14: Add the label for vertical axis.
- Fig.14a: $t^{visu}$ -> $t^{ref}$ ?
- Fig.14b: Add overline if this $t^{ref}$ is an average. Check the same trough the manuscript
- l.245: xtrip -> xstrip
- l.246: smooting -> smoothing
- l.249: for sake of -> for the sake of
- l.251: fiter -> filter
- l.254: What is 'mean'? An average in one cycle?
- Fig.15: Add the label for vertical axis.
- Fig.15: What is '~' on U?
- Fig.15: What is 'brut'?
- Fig.15: Are ●and ○opposite? Check the same through the manuscript
- Fig.15: sliding averaged -> moving average?
- Fig.16: Add the label for vertical axis.
- Fig.16a: What is '~' on U?

- Fig.16b: Does the $ic$ starts from 0? Check the same through the manuscript
- l.260: -8.6$c/U$ -> -8.6?
- l.261: -15$c/U$ -> -15?
- l.261: What is 'chord time'?
- l.261: smothing -> smoothing
- l.263: 3.5$U/c$ -> 3.5$c/U$?

Section 4.2.2
- l.280: As for -> As in or As the same for?
- l.280: separated -> reattachment
- l.282 -2.5$c/U$ -> -2.5
- l.283: What is 'decro'?
- l.284: important?
- l.283-287: These sentences are inconsistent. Which is difficult to detect stall or reattachment?
- l.285: Is the share layer of Fig 9 (h) so close to detect vortex?
- l.285: this?
- l.288: -18$c/U$ -> -18
- Fig.17caption: averaged -> average
- Fig.17a: Add the label for vertical axis.
- Fig.17b: Why are there no data for attached for $ic$=5?

Section 4.2.3
- l.293,296: by(Merius)?
- l.297: averaged -> average
- l.297: Is the 2000 snapshot with 1600Hz during 1.25 sec used for calculation?
- l.298 and Fig.19caption: What is $i$ of $\Psi i$?
- Fig.19: Is the label of color-bar correct?
- Fig.19c: Why the $a(t)$ is plotted in 100Hz not 1600Hz?
- Fig.19caption: averaged -> average

- l.317: $t^{m3}_{stall}$ -> $t^{m3}_{attach}$
- l.317: Do not divide by $U_\infty$.

Section 4.2.4
- l.328: The reason of exploration of instantaneous method is differ from l.244.
- l.347: removed -> subtracted
- l.351: (5)?
- l.353: et?
- Fig.21: Add a label on the vertical axis.
- Fig.22: A label of the vertical axis is not correct.
- Fig.22caption: three methods?
- Fig.22: Is it OK to start by attachment?
- Fig.23: Add a label of the vertical axis
- Fig.23: What is brut?
- Fig.23: Are 'e's different from 'c's?
- Fig.23: Why are there any $sxc/sxcmax$=1?
- Fig.23: Why the value goes to negative while Fig.21is all positive?

Section 5
- l.359: 65 -> 65(4)
- l.360: sensors were not at the trailing edge
- l.362: Why the method of section 4.1 is not included in the method?
- l.368: dicriminate -> discriminate
- l.375: mouvement -> movement
- l.378: strip (electrical) signal?

Reference
- Braud2018: The journal name is mistaken
- Chamorro: N27 -> 27,13, pp1-13
- De Gregorio: The third author's name is mistaken
- Sirovich1987: The journal name is mistaken

---

## Referee Comment (RC2) · Anonymous Referee #2 · 29 Jul 2020

General Remarks: Overall an interesting study, comparing the ETell-Tale derived data with PIV, and employing 3 analytical methods to detect stall and reattachment on a pitching airfoil from PIV data. The authors show interesting insights, but a few points need revision, particularly the phrasing, the notation and details of the experimental and analytical procedures. There are also more than a few typos.

I also note I already read reviewer 1's comments, so I will not repeat them.

Specifically:

Title: ' On the ability of the e-TellTale sensor for the detection of flow stall and reattachment dynamics'

[Figure]

You are thinking about wind turbines, but the measurements are for a straight foil section at small Re. So the present title is a bit misleading, I think. Also, the use of / should be avoided, specially in the title

l11-create>creates l15- smartblade>smart blades l21 - 2007 is not so recent of a reference... l37 - static variations ? l47 - wake width>foil wake (i.e. not the turbine wake)

Fig 2 - show detail of the geometry of the ETellTale sensor, namely the 'pink part'

l68 - what is the mean AOA during the pitching motion ?

Fig 3 - I miss the reference frame

Fig 4 - Rather unclear figure - I miss the reference frame and sx,sy ticks - subfigure a) appears to show a separated flow condition, whereas b) appears attached; is this inconsistently labeled in the caption, or is it because of the camera perspective ? - is 'sy' vertical or along the span ?

l118 - default>a non-identification l120 - So, you corrected the data-points for which the detection algorithm did not work, and set sxc to 1, and these correspond to the blue points in fig 5 ? Why ? It appears the unidentififed data-points occur when the flow is attached. According to fig 5, wouldnt it make sense to 'correct' these data-points to around sxc=0.75 ?

Fig 5 - show y axis variable

Fig 6 - Show airfoil contour

eq (1) - This formula and its application is not so clear ? Use subscripts ? Is ˆthe cross product ? I guess P is a point with coordinates (x,y) and not a vector, and so we cannot define the cross product.

l171 - brutal > massive l184 - you show a characteristic frequency, not the associated time-step

Fig 7 - the a-i points were chosen based solely on visual inspection ? Of how many instantaneous snapshots ? It might explain why you found a consistent lag wrt the analytical stall and reattachment detection methods

Figs 8 & 9 - Show foil contour - use large colorbar font

l207 -you only know the method is robust after you used it...

Fig 11 - larger colorbar font - Remark a2 and a3 are nearly complementary in terms of phase, which is indicative of a succesfull POD

Fig 12 - Use reference "Norberg, C., Fluctuating lift on a circular cylinder: review and new measurements. Journal of Fluids and Structures, 17, pp. 57−96, 2003" To explain slightly higher St for your Re

Fig 15 - what do c=21 and c=9 mean ? mention in the caption - also in the caption >super- imposed

l262 - is this 'dispersion' associated with turbulent structures in shear layers ? Please be specific

l287 - futher>further

l305 - famous DS vortex> Leading edge vortex associated with dynamic stall

Fig 19 - Remark Psi 2 has a much smaller energy content

l313 - since it appears all stall detection methods are early wrt the visual reference, does it make sense to adjust the visual reference ?

Tables 1 & 2 - Use more concise column titles !!

l336 - Ability of the e-TellTale sensor to detect flow stall

Fig 23 - the fluctuations on the 'brut' signal appear to be of the same frequency as the sampling you are showing... and some of these higher frequencies are still seen with a moving average e=9.

Do you know what is the natural frequency of the tell-tale sensor ? This might have a crucial influence, and MUST be considered, for both laboratory and field experiment design

l363 - trailing edge > in the aft region

---

## Author Comment (AC1) · 13 Oct 2020

Anonymous referee #1 General comments

Q1 : The importance of this work lies on the evaluation for e-TellTale but not for a tuft. It should be explained if there are any difficulties specific for e-TellTale to follow the flow dynamics, or to be recognized by image processing conducted in this work.

The most important feature of the sensor is the electrical sensing. But the electrical signals were not evaluated in this work. The correlation of the signals to the strip position should be described more in detail especially if there are some issues left.

[Figure]

A1 : We agree with the reviewer, this work was firstly conducted with full e-TellTale with the electronic sensing, unfortunately, the strain gauge sensor was damaged by the laser sheet during the first tests. To make it clearer to the reader, this was changed in the title. "Low Reynolds investigations on the ability of the strip of e-Telltale sensor to detect flow features over wind turbine blade section: flow stall and reattachment dynamics" instead of "Ability of the strip of e-TellTale sensor to detect flow features over wind turbine blades: flow stall/reattachment dynamics" Therefore the rest of the experiments were performed without the electronic sensing and with a nylon strip. However working with the strip provide a lot of information which was a great help for the development of a future scaled down functional e-TellTale. Preliminary tests (without records unfortunately) were performed before the e-Telltale damage to check that the down scaled e-Telltale signal has a similar qualitative behavior than for the full size e-Telltale has explained in the article L75: "The signal from the strain gauge sensor was not acquired simultaneously during PIV measurements, however, it has been checked before experiments that the signal from this strip, made of a nylon fabric, behaves similarly as full-scale experiments from (Soulier et al., 2017). In particular it was checked that it was possible to distinguish two levels of the signal within the blade oscillation cycle, corresponding to two different flow states over the aerodynamic surface: attached at least at the leading edge/stalled."

To make it clearer, the reference to figure 1b has been added: "The signal from the strain gauge sensor was not acquired simultaneously during PIV measurements, however, it has been checked before experiments that the signal from this strip, made of a nylon fabric, behaves similarly as full-scale experiments from (Soulier et al., 2017). In particular it was checked that it was possible to distinguish two levels of the signal within the blade oscillation cycle, corresponding to two different flow states over the aerodynamic surface: attached/stalled (see figure 1b)."

The correlation between the position of the strip and the signal have not been registered in this study and the will be done in some future studies

Finally the scope of the paper is to demonstrate that the strip is following the flow with regard to the separation and stall aerodynamic properties. As explained in the conclusion: what is missing now is the relation between the strip and the strain gauge signal.

Q2 : If the authors intended to scale-down the full-scale device, the way of design to scale-down should be explained. The experimental condition or the configuration of the sensor for the full-scale wind tunnel test is not clear because the cited reference seems not yet published.

A2 : The down-scaling of the e-Telltale signal was made with the intention to reproduce the main characteristics of the full scale e-Telltale signal which are: - first rise of the e-Telltale signal at the trailing edge separation angle - sudden increase of the e-Telltale signal at the stall angle These tests were performed prior to PIV measurements from visualisation of the strain gauge signal and using wool tuft distributed on the suction side of the blade (for a fast evaluation of the trailing edge separation angle and the stall angle). Indeed, full scale experiments are not yet published, we reported the important properties needed for the present article in figure 1, which presents the e-TellTale signal first rise and sudden increase.

Q3 : The TR-PIV is conducted in 2D. Does the 3D motion affect the electrical signals? To think about this, it is recommended to describe more about the configuration of the e-TellTale in detail including the 'stainless sheet' and the 'small part'.

A3 : As explained in A1, the scope of the paper is the strip motion, not the electronic signal, which was not registered. The full scale measurements, that will be published soon, show an increase of the variance for stall angles, which may be related to what is observed on the downscale strip using PIV (out of plan motions of the strip). However, this should be confirmed with e-TellTale electronic signal.

Q4 : The position detection is the most important technique in this work. To ensure the validity of the experiment, clear and correct explanation is necessary. For example, why

sx replaced to sxmax instead of sxmin for the state beyond the stall in Fig.15 while sx is decreasing when the flow is detached according to Fig.4.? Is the sx really reaches to 0 at around 0.9s and 5.0s as shown in Fig.15 while the length of the strip is only 0.3c?

A4 : The inconsistencies pointed out by the reviewer are due to the merger of different versions of the manuscript. Indeed the choice for the direction of sx was changed during the writing of the article some old figures/errors have not been corrected yet, It has been corrected in the article (Section 3.1 and Figure 4). Fig 15 is not about sx, but at the end of the article the Fig 21 deals with sx and was already right. sx/smax= 0 does not correspond to the strip at the leading edge. There is here a shortcut that is misleading the reader: sx/sxmax in figure 4. is expressed with the origin of the coordinate system placed at a position at which the minimum of sx is 0. To make it clearer to the reader, it has been modified as follow: - sxp (in pixels in a coordinates system corresponding to the image sides) variable has been introduced and the change of coordinate system is explained

Q5 : The objective and the result of the three postprocessing analysis is not clear.The discussion about these analyses is too long and confusing while this manuscript is worthwhile enough for publishing even without these analyses.

A5 : From PIV measurements there are many methods developed to detect the flow separation. But none of them were compared with each others. Moreover, these existing methods were adapted in the present paper. Because the purpose of the paper is an evaluation of the strip to detect flow separation, it was found necessary to have a first assessment of the developed detection methods. This have been modified in the article to make it clearer: "To be able to study the ability of the strip to detect the instants of the flow stall/reattachment phenomena it was necessary to use methods allowing to detect these flow characteristics from PIV velocity fields. Several methods were identified from the literature but never compared and not completely adapted to our needs They were adapted here and compared between each others, providing a first assessment of the methods before comparisons with the strip movements." instead of "To be

able to study the ability of the strip to detect the instants of the flow stall/reattachment phenomena, three robust detection methods were applied to the flow field obtained from the TR-PIV measurements"

Q6 : 'Because the definition of stall and reattachment instants is a complex problem' at l.321 is not clear to understand the objective because 'the definition' shown in section 4.1 is not complex.

A6 : Although the description of stall given in section 4.1 is relatively straightforward, in practice it may be difficult to identify the onset of stall and reattachment directly from PIV measurements. Assessing the relevance of different identification methods that are relatively new is useful. In order to do this, we compared four different methods. We agree with the reviewer, the world complex is not suitable. The sentence has been modified as follow: "From PIV measurements there is no unique criteria to detect stall instants."

Q7 : If the objective of the analysis is to investigate the local flow phenomena which governs the motion of the strip, you might mention something more from the small l/c results of the method 1.

If the objective of the analysis is to evaluate the accuracy of each methods to detect the instants, the parameters for each method (such as x=c or l=c for the method 1) should be optimized before the comparison.

A7 : l/c has firstly been chosen to be of the order on magnitude of the mean recirculation width in the normal direction. Different values of l/c in the range [0.07, 0.7] were considered in order to check for possible dependence of the detection instants. To make it clearer to the reader, this is now explained in the article Since the purpose of the present article is to evaluate the detected instants at the strip location, so to avoid any possible delays, the chosen x/c is the location of the strip. No exploration of x/c were performed, which is out of the scope of the paper.

[Figure]

Q8 : In section 5, there are no explanation that the exact instants tref was defined by the visualization of the velocity field. Moreover, it is concluded that the strip capabilities to follow the stall/reattachment dynamics was validated by comparison to the three methods while the most direct validation seems to come from the comparison to tref . These are very confusing.

A8 : Tref definition is present in the section 4.1. We do use as a reference a simple visual technique, however it may not always be possible to rely on such approaches, in particular in the case of very large data-sets, which is why we consider different methods.

Q9 : The validity of the zero-crossing criteria is not clear. For about the 'resolution', describe the way of evaluation of 3.5c/U at l.262. Clarify the meaning of the phrase 'at the limit of the measurement precision' in l.265.

A9 : 3.5c/U is the dimensionless temporal resolution from the PIV sampling frequency, i.e the time between two PIV flow fields (corresponding to a physical separation of 0,01s as the sampling frequency is 100Hz). This has been added to the revised version of the manuscript.

Q10 : It should be described if there are reasons to set the detection threshold as zero. I think it should be optimized for each stall/reattachment instants for each method. Maybe this causes the 'bias' in l.350. Ideally, those instants should be compared to tref after the optimization.

A10 : The intended objective of the chosen threshold (zero crossing method) is to be able to have a way to compare detection methods with each others. Moreover, changing the threshold value won't bring a universal threshold value to use in other datasets whereas using the zero-crossing method can be used anywhere without arbitrary values. It is true that we could try to find an optimal criterion for each method. However, it is not true that :I) there is some arbitrariness in the fact that we are choosing the mean as a threshold, II) the results could depend on the value of the sampled mean,

as the signals are not exactly cyclic. Regarding I), The criterion – zero-crossing – has the advantage that it is the same for all methods, and that it does not depend on the signal intensity or on the particular cycle considered. Regarding ii) we show that the results presented in the paper, which were obtained using the average over the full signal length (18 cycles), were not significantly modified when the average was taken only over a small fraction of the signal length (corresponding to the first few cycles). The chosen criterion therefore appears to be both universal and robust.

Q11 : Moreover, if zero is calculated using the mean value in one cycle, the strategy on how to apply this to the field should be explained because the motion is not cyclic in the field.

A11 : It is true that the motions are not entirely cyclic. In the paper, the mean value was taken over the full signal length (all cycles). However we found that a good estimate for this value was obtained by taking the average of the signal over only a few cycles (for instance the first three or four), so that the separation and reattachment onset times were not significantly modified.

Q12 : The delay of the reattachment instances is described to be owing to the smoothing procedure in many sections. But I think the reason lies not only in the smoothing procedure but also in this threshold setting.

A12 : We agree with the reviewer, that the times could be influenced by several factors, such as the threshold (mean) value and the smoothing procedure. However, preliminary sensitivity analysis suggested that the length of the moving average window used reduces the noise which has a stronger effect on the detection times than the small variations in the sampled mean value of the signal, which is relatively well approximated with only a few cycles. To be clearer to the reader, the following sentence has been modified : "The main bias of this smoothing procedure is to reduce the slope during the change of flow state as illustrated in figure 15 and because of the modification of the slope, a constant bias is introduced in the detected instants. Another bias can also be

introduced due to the chosen threshold value (zero-crossing method) However, filter size as high as 21 time steps were found necessary to have an automatic procedure to extract stall and reattachment instants for all detection methods and thus having comparable results. In that case, the most important bias on the detected instants comes from the smoothing procedure. instead of : "The main bias of this smoothing procedure is to reduce the slope as illustrated in figure 15. Larger filter size have a larger impact on the gradients, however, filter size as high as 21 time steps were found necessary to have an automatic procedure to extract stall and reattachment instants for all detection methods and thus having comparable results"

Q13 : To think more about the interesting results that the dispersion of the delay is larger for reattachment than for stall, showing the average and the dispersion of the (td-tc) and the (th-tg) not only (tc+td) and (tg+th) is recommended to understand the rapidity of each phenomena.

A13 : We agree that this would be an interesting investigation, however we would like to emphasize that the acquisition frequency of the present results is 100Hz, so that the time resolution is too small for that purpose.

---

## Author Comment (AC2) · 13 Oct 2020

The technical comments will be taken into account directly in the article

Anonymous referee #2

The majority of the comments will be taken into account directly in the article

Here are a few complementary answers:

Q14 : You are thinking about wind turbines, but the measurements are for a straight foil section at small Re. So the present title is a bit misleading, I think. Also, the use of / should be avoided, specially in the title A14 : We agree with the reviewer, the title

has been modified as follows: " Low Reynolds investigations on the ability of the strip of e-TellTale sensor to detect flow features over wind turbine blade section: flow stall and reattachment dynamics"

Q15 : -l68 - what is the mean AOA during the pitching motion ?

A15 : We only have a relative measurement of the AoA (from PIV measurements). Prior to PIV acquisitions; wool tuft were placed chordwise and the AoA amplitude was chosen to include the stall angle.

Q16 : l118 - default>a non-identification l120 - So, you corrected the data-points for which the detection algorithm did not work, and set sxc to 1, and these correspond to the bluepoints in fig 5 ? Why ? It appears the unidentififed data-points occur when the flow is attached. According to fig 5, wouldnt it make sense to 'correct' these data-points to around sxc=0.75 ? A16: We have modified the sentence to be more clear. During stall the e-telltale may leave the field of view hence missing data points. To estimate the missing data, we use a crude estimation method, which is to replace the missing points with an arbitrary value (taken here to be equal to the minimum value encountered). It is likely that using the measured average value over the stall as suggested by the reviewer, would provide a better estimation of the missing values. However, tests for which this average value was replaced did not lead to significant changes in the detection times. The sentence has been replaced as follows: "During stall there is a significant amount of out of plan motions of the strip from the laser sheet. In those cases, the strip was not enlightened inducing missing values in sxp as can be seen in the figure 4c. These values were replaced by the maximum value of sx . The corrected signal, sxc , is presented with the original signal sx in the figure 5." instead of : "Missing values present in the signal are related to default in the contour detection algorithm as can be seen in the figure 4c. These outliers are found to be correlated with AoA beyond stall, were 3D effects are dominants. These values were replaced by the minimum value of sx . The corrected signal, sxc , is presented with the original signal sx in the figure 5."

Q17 : -Fig 7 - the a-i points were chosen based solely on visual inspection ? Of how many instantaneous snapshots ? It might explain why you found a consistent lag wrt the analytical stall and reattachment detection methods

A17 : Yes it is based only on visual inspection, on 2000 snapshots. The determination of the detection instants is likely to be affected by different factors, such as the exact threshold (signal mean) value, the length of the moving average window in the smoothing procedure, and the estimation procedure for missing data in the case of the strip contour detection method. It is therefore difficult to provide a reliable interpretation of the small differences observed between the different methods.

Q18 : l262 - is this 'dispersion' associated with turbulent structures in shear layers ? Please be specific

A18 : This dispersion is related to the variance of the signal. There is certainly a link with the turbulent structures in the shear layers, however the time resolution in this study is not enough high to investigate this point further and need a dedicated work which is out of the scope of the present paper.

Q19 : l313 - since it appears all stall detection methods are early wrt the visual reference,does it make sense to adjust the visual reference ?

A19 : As already pointed out in A17, the delay on the detected instants from the 3 detection methods is quantified relatively to the threshold and the smoothing method. It is true that it would make sense to adjust the visual reference. However we have not been able to determine a criterion that would allow us to do so a priori.

Q20 : Fig 23 - the fluctuations on the 'brut' signal appear to be of the same frequency as the sampling you are showing...and some of these higher frequencies are still seen with a moving average e=9.C3 Do you know what is the natural frequency of the tell-tale sensor ? This might have a crucial influence, and MUST be considered, for both laboratory and field experiment design

A20 : We agree with the reviewer that the time resolution of the signal is not enough to study these oscillations. Regarding the resonance of the strip, the material used has a resonance frequency that can't be extracted using standard methods (indentation, traction/compression ...). However, this natural frequency has been observed prior to measurements and avoided in these tests using another free stream velocity. This phenomena needs a complete characterization by itself that is out of the scope of the present paper.